# Daedalus Ionospheric Profile Continuation (DIPCont): Monte Carlo Studies Assessing the Quality of In Situ Measurement Extrapolation

Joachim Vogt[1,7], Octav Marghitu[2], Adrian Blagau[2,1,7], Leonie Pick[3,1,7], Nele Stachlys[4,1,7],
Stephan Buchert[5], Theodoros Sarris[6], Stelios Tourgaidis[6], Thanasis Balafoutis[6], Dimitrios Baloukidis[6],
and Panagiotis Pirnaris[6]

[1]School of Science, Constructor University, Campus Ring, 28759 Bremen, Germany
[2]Institute for Space Science, Str. Atomistilor 409, Ro 077125, Bucharest-Magurele, Romania
[3]Institute for Solar-Terrestrial Physics, German Aerospace Center, Kalkhorstweg 53, 17235 Neustrelitz, Germany
[4]Leibniz Institute for Astrophysics Potsdam (AIP), An der Sternwarte 16, 14482 Potsdam, Germany
[5]Swedish Institute of Space Physics, Uppsala, 75121, Sweden
[6]Department of Electrical and Computer Engineering, Democritus University of Thrace, Xanthi, 67132, Greece
[7]Until December 2022, Constructor University operated under the name Jacobs University Bremen

**Correspondence:** Joachim Vogt (jvogt@constructor.university)

**Abstract.** In situ satellite exploration of the lower thermosphere and ionosphere (LTI) as anticipated in the recent Daedalus mission proposal to ESA will be essential to advance the understanding of the interface between the Earth's atmosphere and its space environment. To address physical processes also below perigee, in situ measurements are to be extrapolated using models of the LTI. Motivated by the need for assessing how cost-critical mission elements such as perigee and apogee distances as well as the number of spacecraft affect the accuracy of scientific inference in the LTI, the Daedalus Ionospheric Profile Continuation (DIPCont) project is concerned with the attainable quality of in situ measurement extrapolation for different mission parameters and configurations. This report introduces the methodological framework of the DIPCont approach. Once a LTI model is chosen, ensembles of model parameters are created by means of Monte Carlo simulations using synthetic measurements based on model predictions and relative uncertainties as specified in the Daedalus Report for Assessment. The parameter ensembles give rise to ensembles of model altitude profiles for LTI variables of interest. Extrapolation quality is quantified by statistics derived from the altitude profile ensembles. The vertical extent of meaningful profile continuation is captured by the concept of extrapolation horizons defined as the boundaries of regions where the deviations remain below a prescribed error threshold. To demonstrate the methodology, the initial version of the DIPCont package presented in this paper contains a simplified LTI model with a small number of parameters. As a major source of variability, the pronounced change of temperature across the LTI is captured by self-consistent non-isothermal neutral density and electron density profiles, constructed from scale height profiles that increase linearly with altitude. The resulting extrapolation horizons are presented for dual-satellite measurements at different inter-spacecraft distances but also for the single-satellite case to compare the two basic mission scenarios under consideration. DIPCont models and procedures are implemented in a collection of Python modules and Jupyter notebooks supplementing this report.

# 1 Introduction

The lower thermosphere and ionosphere (LTI) at altitudes between about 100 km and 200 km is characterized by transitions of several atmospheric attributes. It is the lower part of the heterosphere where atmospheric constituents are no longer mixed by turbulence, and start to follow separate barometric laws (e.g., Picone et al., 2002; Izakov, 2007). As part of the thermosphere, the temperature profile shows a significant increase with altitude throughout the whole LTI (e.g., Chamberlain and Hunten, 1987). As part of the ionosphere, it includes the E-layer peak in electron density and the bottom side of the F-layer (e.g., Hargreaves, 1992). With strongly altitude-dependent neutral-ion and neutral-electron collision frequencies, the LTI supports an anisotropic conductivity tensor that gives rise to a complex interplay of electric fields and currents. The conductivity tensor components affecting the directions perpendicular to the ambient magnetic field, namely, Pedersen and Hall conductivities, show pronounced maxima in the LTI (e.g., Baumjohann and Treumann, 1996). A key variable quantifying its energetics is the Joule heating rate. Particular rich dynamics can be observed in the auroral region at high latitudes where energy and momentum from the magnetosphere are fed into the ionosphere through currents flowing parallel to the ambient magnetic field lines (e.g., Vogt et al., 1999). A comprehensive review of LTI features, measurement techniques, and models is provided by Palmroth et al. (2021).

Since the early 20th century, the LTI has been studied extensively using ground-based remote sensing facilities such as ionosondes and radars, but in all aspects requiring in situ observations it remains underexplored territory. Rocket flights (e.g., Sangalli et al., 2009; Pfaff et al., 2022a) can offer only local and temporally confined information. Major technical challenges have so far prevented a satellite mission to the deep, dense part of the LTI, despite scientific interest, community proposals, and feasibility studies by major space agencies (e.g., Grebowsky and Gervin, 2001; Pfaff et al., 2022b). An early conception of the TIMED mission (e.g., Yee et al., 1999) considered dipper options for in situ investigations of the LTI. A recent initiative along this line is the Daedalus mission proposal (Sarris et al., 2020), submitted to ESA in response to the Explorer 10 Call under the Earth Observation Program, and selected together with two other proposals for a Phase-0 science and technical study. Daedalus aims to perform in situ measurements in the LTI from an elliptical orbit, with a nominal perigee of 150 km and an apogee on the order of 2000 km. Very low altitudes down to 120 km will be sampled by use of propulsion, through a series of short excursions in the form of perigee descent maneuvers. These are planned to be performed at high latitudes (>65 degrees magnetic latitude), where Pedersen conductivity and Joule heating maximize. The highly elliptical orbit of Daedalus leads to a natural precession of the orbit's semi major axis, both in magnetic latitude and in magnetic local time; this means that Daedalus will perform measurements along its elliptical orbit down to the nominal perigee of 150 km throughout all magnetic latitudes. The geophysical observables sampled by Daedalus will enable obtaining a series of derived products, as described in Table 1 of the Daedalus Report for Assessment (ESA, 2020), which, among many others, include the calculation of Pedersen conductivity and Hall conductivity.

The range of accessible perigees will be particular critical for any future LTI satellite mission, with severe impact on the propellant budget and other mission performance parameters (Sarris et al., 2020). With nominal perigees not much less than 150 km, peak conductivities and currents controlling E-region electrodynamics lie typically below the orbits. Physically mean-

ingful downward continuation of in situ satellite measurements is desired, ideally using state-of-the-art models of the LTI (e.g., Sarris et al., 2023b). Another critical element of LTI mission conception is the number of spacecraft (Sarris et al., 2023a). The *Daedalus Ionospheric Profile Continuation (DIPCont)* project is concerned with vertical profiles of LTI variables and their reconstruction from dual-spacecraft and single-spacecraft observations. More specifically, the focus of the project is on the *quality* of profile continuation towards the lower LTI with its maxima in conductivities and current intensities, as given by the accuracy, the resolution, and the coverage of the reconstructions obtained from in situ measurements. Inspired by early work to extrapolate vertical profiles carried out under the Daedalus Phase-0 Science Study, DIPCont introduces a systematic probabilistic approach to the problem.

The DIPCont procedure to assess the quality of in situ measurement downward continuation is detailed in Section 3. In brief, after choosing a LTI model, representative ensembles of altitude profiles are generated by means of Monte Carlo simulations. The altitude profile ensembles give rise to statistical measures of relative deviation which in turn allow for estimating extrapolation horizons, effectively capturing the altitude range where deviations remain within given error thresholds. The basic ideas are illustrated in Figure 1, displaying electron density and Pedersen conductivity extrapolation horizons for a range of relative error thresholds along the orbits of a dual-satellite mission. Horizontal distance corresponds to the latitudinal (north-south) direction, with the origin of the horizontal axis centered at the highest latitude along the satellite orbits. In the LTI model runs leading to Figure 1, latitudinal inhomogeneity parameters are set to reproduce the two electron density maxima observed by a polar orbiting satellite when crossing the auroral oval. See Section 4 and Appendix D for details. It is important to note that the filled contour representations of electron density and Pedersen conductivity model distributions mainly serve to provide contextual information, while the essential results of the DIPCont modeling procedure are the extrapolation horizons represented as plain contour lines, in response to the satellite orbit configuration (white lines). The extrapolation horizons of the model run shown in Figure 1 suggest that for a dual-satellite mission as anticipated in the Daedalus Report for Assessment (ESA, 2020), downward continuation yields relative errors of a few ten percent at altitudes where electron density and Pedersen conductivity maximizes. Implications are discussed in more detail further below in Section 4, and contrasted with the single-satellite case.

The LTI model used to introduce and demonstrate the DIPCont methodology in this paper is presented in Section 2. The parametric model captures the whole LTI temperature range and thus addresses a main source of variability. To limit the number of model parameters and thus also instabilities during model inversion in this initial DIPCont study, LTI variables showing less pronounced changes and ionization source mechanisms are treated in a simplified manner. Furthermore, since the quality of downward continuation is in the focus of our study, the LTI model is restricted to E-region physics, with the influence of the F-region left for future work.

Further first results are presented in Section 4, including a brief comparison between the single-spacecraft and the dual-spacecraft scenario. In Section 5, our findings are discussed in the context of important technical parameters and constraints relevant for a low-altitude mission. The body of the paper is concluded in Section 6 with prospects for upcoming work. Model derivations and technical details are presented in the Appendices, with particular emphasis on the incorporation of a non-isothermal temperature profile varying linearly with altitude.

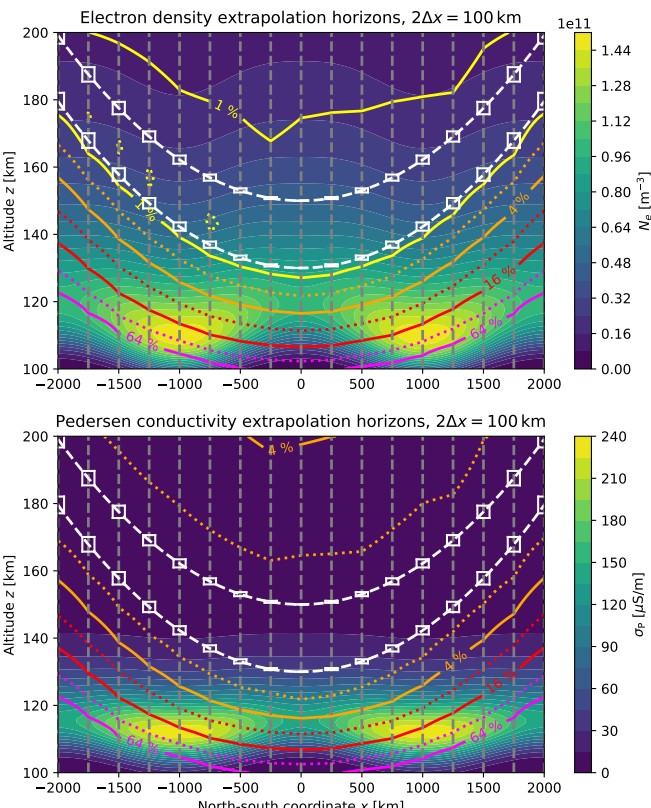

**Figure 1.** Extrapolation horizons and orbit configuration displayed on top of a two-dimensional section of the modeled LTI. Upper panel: Electron density $N_e$. Lower panel: Pedersen conductivity $\sigma_P$. Horizontal distance corresponds to the latitudinal (north-south) direction, with the origin of the horizontal axis centered at the highest latitude along the satellite orbits. In the LTI model runs leading to this figure, latitudinal inhomogeneity parameters are set to reproduce the two electron density maxima observed by a polar orbiting satellite when crossing the auroral oval. Synthetic measurements are produced along the two satellite orbits (white dashed lines). The parameters of vertical profiles are estimated using measurements within a window (white solid rectangle) of width $2\Delta x$ around the nodes of a horizontal grid (gray dashed lines). Extrapolation horizons (solid and dotted colored lines) for a set of relative error levels are displayed as contours of a relative deviation measure, here the root-mean-square deviation of the ensemble of extrapolated profiles from the synthetic model prediction.

## 2 Parametric models of LTI variables

Probabilistic measures of extrapolation quality produced by the DIPCont procedure detailed in Section 3 are based on synthetic in situ observations predicted by a model of the LTI. As emphasized in space physics textbooks and reviews of the LTI (e.g., Pfaff, 2012; Richmond, 1995), the full complexity of LTI variability and dynamics calls for a full multi-species description, taking into account source and loss processes varying in importance and efficiency as functions of magnetic latitude and local time and further factors. In the future, DIPCont functionality is planned to be included in the Daedalus MASE (Mission Assessment through Simulation Exercise) toolset (Sarris et al., 2023b), designed with the purpose to assess and demonstrate the closure of the mission objectives of the proposed Daedalus mission.

The more complex the LTI model of choice, however, the larger the number of parameters that are to be estimated with a downward continuation of in situ satellite measurements, which in turn tend to negatively affect the stability of model inversion. With these implications in mind, the initial version of the DIPCont package contains a simplified LTI description based on a limited set of parameters. Extrapolation quality of a single but important process, namely, the formation of Pedersen conductivity $\sigma_P$, is supposed to be studied in a self-consistent manner. To this end, only a single particle species is considered, and classical photoionization physics is applied to parametrize ionospheric layer formation. Furthermore, as explained in reviews of ionospheric physics (e.g., Rishbeth, 1997), contributions from electron-neutral collisions to the Pedersen conductivity $\sigma_P$ peak in the D-region but are unimportant at higher altitudes, see also Figure 4 in Sarris et al. (2023b). We thus arrive at the expression

$$\sigma_P = \frac{N_e e^2}{m_i} \frac{\nu_{in}}{\nu_{in}^2 + \Omega_i^2} \tag{1}$$

($e$: elementary charge, $m_i$: ion particle mass, $\Omega_i$: ion gyrofrequency), suggesting that the altitude variabilities of electron number density $N_e$ and ion-neutral collision frequency $\nu_{in}$ need to be modeled carefully. Less critical is the dependence of ion gyrofrequency $\Omega_i$ on magnetic field strength as it does not vary much over the LTI altitude range, and is captured with sufficient accuracy by well-established empirical models. Different parametrizations exist for the ion-neutral collision frequency $\nu_{in}$ (e.g., Palmroth et al., 2021; Huba, 2019; Evans et al., 1977). In general the expressions are directly proportional to the number density $N_n$ of neutral particles. As presented in the Appendices A and B, also the self-consistent construction of electron density $N_e$ rests prominently on the $N_n$ profile, which in turn is conveniently modeled in terms of the density scale height $H_n^N$. This aspect is chosen as a starting point below in Subsection 2.1, to further explain and motivate the LTI modeling approach.

The LTI model can be summarized in the form $\mathbf{m} = \mathbf{m}(z|\mathbf{p})$ with a vector $\mathbf{m}$ of LTI observables and derived functions, and a vector $\mathbf{p} = \mathbf{p}(x)$ of model parameters, separating the primary (strong) dependence on altitude $z$ from the secondary (weak) dependence on horizontal location $x$ in a numerically efficient manner. Note the model functions are *local representations* of altitude profiles in the sense that they refer to a flexible reference altitude, $z_0$, that can be adapted to the locations where measurements are taken. In the DIPCont development phase it was observed that parameters of model functions in local representations typically showed weaker correlations and could be estimated more reliably, in particular as compared to the *regional representations*, relying on parameters at some fixed altitude, like the peak electron density height.

As in the Daedalus Report for Assessment (ESA, 2020), the vertical boundaries of the LTI region are assumed to be at $z_\mathrm{B} = 100\,\mathrm{km}$ (base or bottomside altitude) and $z_\mathrm{T} = 200\,\mathrm{km}$ (topside altitude).

## 2.1 Scale height parameters

As demonstrated in Appendix A, profiles of (neutral gas) pressure $P_n$ and neutral (number) density $N_n$ are conveniently constructed using

$$P_n(z) = P_{n0} \exp\left\{ -\int_{z_0}^{z} \frac{\mathrm{d}\tilde{z}}{H_n^P(\tilde{z})} \right\} , \tag{2}$$

$$N_n(z) = N_{n0} \exp\left\{ -\int_{z_0}^{z} \frac{\mathrm{d}\tilde{z}}{H_n^N(\tilde{z})} \right\} , \tag{3}$$

where $H_n^P$ and $H_n^N$ denote the pressure scale height and the density scale height, respectively. Furthermore, it is shown that

$$H_n^N = H_n^P \left( 1 + \frac{\mathrm{d}H_n^P}{\mathrm{d}z} \right)^{-1} \tag{4}$$

if the pressure scale height

$$H_n^P = \frac{kT_n}{m_n g} = \frac{R_\mathrm{gas} T_n}{M_n g} \tag{5}$$

($k$: Boltzmann constant, $T_n$: neutral temperature, $g$: gravitational acceleration, $R_\mathrm{gas}$: universal gas constant, $m_n$: average particle mass, $M_n$: average molar mass) changes only with temperature $T_n$. Eqs. (4) and (5) further imply that, if temperature $T_n$

varies linearly with altitude $z$, then also $H_n^P$ and $H_n^N$. In Appendix A it is shown that the constant inverse gradients

$$\gamma = \left( \frac{\mathrm{d}H_n^P}{\mathrm{d}z} \right)^{-1} \tag{6}$$

and

$$\eta = \left( \frac{\mathrm{d}H_n^N}{\mathrm{d}z} \right)^{-1} \tag{7}$$

are related through

$$\eta = \gamma + 1 . \tag{8}$$

Variations of gravity $g$ across the LTI are in the range of a few percent and can be neglected in this context. Profiles of $T_n$, $M_n$, and $H_n^P$ as predicted by the empirical atmospheric model NRLMSIS 2.0 (Emmert et al., 2021) for different seasons and latitudes are displayed in Figures S1a–S1d as part of the supplementary material to this paper, indicating that relative variations of average molar mass are indeed significantly smaller than those of neutral temperature. We thus disregard altitude changes in

average molar mass $M_n$ as imposed by changes in atmospheric composition, and further assume that temperature $T_n$, pressure scale height $H_n^P$, and density $H_n^N$ vary linearly with altitude in a self-consistent manner as described by Eqs. (4) and (5).

According to Eq. (A11), the local density scale height $H_{n0}^N$ can be obtained using the inverse scale height gradients and the local pressure scale height $H_{n0}^N$ from the expression

$$
\begin{aligned}
H_{n0}^N &= \frac{H_{n0}^P}{1 + \gamma^{-1}} = \frac{\gamma}{\gamma + 1} H_{n0}^P = \frac{\eta - 1}{\eta} H_{n0}^P \\
&= \frac{\eta - 1}{\eta} \frac{R_{\text{gas}} T_{n0}}{M_n g} \,,
\end{aligned}
\tag{9}
$$

where the subscript 0 indicates that the respective variable is taken at the reference (measurement) altitude $z_0$.

Since in the construction of neutral and electron density profiles (Appendices A and B) the scale height parameters are of central importance, their relative change is reflected in the following reference values: at $z_{\text{B}}$, $T_{n\text{B}} \sim 200\,\text{K}$ and $H_{n\text{B}}^P \sim 6\,\text{km}$. Supplementary Figures S1a–S1d suggest that the pressure scale height varies by a factor $\sim 5$ across the LTI, thus $H_{n\text{T}}^P \sim 5 \cdot H_{n\text{B}}^P \sim 30\,\text{km}$ and $T_{n\text{T}} \sim 1000\,\text{K}$ at $z_{\text{T}} = 200\,\text{km}$. We obtain $\mathrm{d}H_n^P/\mathrm{d}z = 0.24 = \gamma^{-1}$, $\gamma \sim 4$, $\eta = \gamma + 1 \sim 5$, and $H_{n\text{B}}^N = \frac{\eta-1}{\eta} H_{n\text{B}}^P \sim 5\,\text{km}$ for the density scale height at the base of the LTI.

## 2.2 Neutral temperature

Neutral temperature $T_n$ is assumed to vary linearly with altitude $z$:

$$
T_n(z|z_0, T_{n0}, L_{n0}) = T_{n0} \cdot \left(1 + \frac{z - z_0}{L_{n0}}\right) \,.
\tag{10}
$$

The parameters $T_{n0}$ and $L_{n0}$ are the neutral temperature and the gradient length scale, respectively, at a reference altitude $z_0$. The constant temperature gradient is given by

$$
\frac{\mathrm{d}T_n}{\mathrm{d}z} = \frac{T_{n0}}{L_{n0}} \,.
\tag{11}
$$

Using Eq. (A13), the neutral temperature profile can be expressed by means of the parameters $\eta$ and $H_{n0}^N$ as follows:

$$
T_n(z|z_0, T_{n0}, H_{n0}^N, \eta) = T_{n0} \cdot \left(1 + \frac{z - z_0}{\eta H_{n0}^N}\right) \,.
\tag{12}
$$

## 2.3 Neutral density

The altitude dependence of neutral density $N_n$ for linear scale height profiles is derived in Appendix A, resulting in the following local representation

$$
\begin{aligned}
&N_n(z|z_0, N_{n0}, H_{n0}^N, \eta) \\
&= N_{n0} \cdot \exp\left\{-\eta \ln\left(1 + \frac{z - z_0}{\eta H_{n0}^N}\right)\right\} \\
&= N_{n0} \cdot \left(1 + \frac{z - z_0}{\eta H_{n0}^N}\right)^{-\eta} \,,
\end{aligned}
\tag{13}
$$

see also Eq. (A15). The parameter $N_{n0} = N_n(z_0)$ is the local neutral density, i.e., its value at the reference altitude $z_0$.

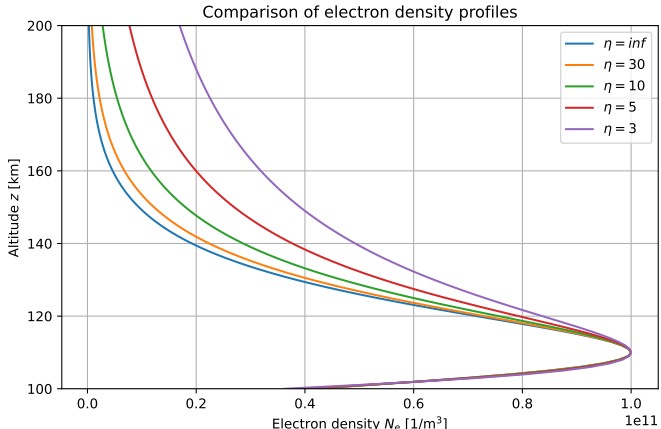

**Figure 2.** Altitude dependence of the non-isothermal electron density model for different values of the inverse neutral density scale height gradient $\eta$. Common electron density peak parameters are $z_* = 110\,\mathrm{km}$, $N_{e*} = 10^{11}\,\mathrm{m}^{-3}$, $H_{n*}^N = 7\,\mathrm{km}$. The case $\eta \to \infty$ (in the legend, $\eta = \mathrm{inf}$) corresponds to the isothermal limit.

## 2.4 Electron density

The altitude dependence of electron density $N_e$ for linear scale height profiles is derived in Appendix B, resulting in the following local representation

$$
\begin{aligned}
&N_e(z|z_0, N_{e0}, L_{r0}\cos\chi, H_{n0}^N, \eta) \\
&= N_{e0}\exp\left\{\tfrac{1}{2}\tfrac{\eta}{\eta-1}\left[-\theta_0 + \tfrac{H_{n0}^N}{L_{r0}\cos\chi}\left(1 - e^{-\theta_0}\right)\right]\right\}
\end{aligned}
\tag{14}
$$

with $\theta_0 = \theta_0(z) = (\eta-1)\ln\left(1 + \tfrac{z-z_0}{\eta H_{n0}^N}\right)$, see Eq. (B16). The parameter $N_{e0} = N_e(z_0)$ gives electron density at the chosen reference altitude $z_0$. Note that $L_{r0}$ and $\chi$, the angle of incident radiation with the atmospheric layer normal direction, cannot be estimated separately but only combined as $L_{r0}\cos\chi$. The parameters $H_{n0}^N$ and $\eta$ can be inherited from estimations using neutral temperature and/or neutral density data, effectively reducing the number of electron density parameters and thus stabilizing the estimation procedure.

The non-isothermal electron density model can also be expressed in terms of the ionization peak parameters, namely, the altitude $z_*$ and the electron density value $N_{e*} = N_e(z_*)$:

$$
\begin{aligned}
&N_e(z|z_*, N_{e*}, H_{n*}^N, \eta) \\
&= N_{e*}\exp\left\{\tfrac{1}{2}\tfrac{\eta}{\eta-1}\left[-\theta_* + 1 - e^{-\theta_*}\right]\right\},
\end{aligned}
\tag{15}
$$

with $\theta_* = \theta_*(z) = (\eta-1)\ln\left(1 + \tfrac{z-z_*}{\eta H_{n*}^N}\right)$, and $H_{n*}^N$ denoting the density scale height at $z = z_*$. See Appendix B2 for details. Electron density profiles for identical peak parameters but different values of $\eta$ are displayed in Figure 2.

The electron density model is designed to describe the ionospheric E-layer, assuming that contributions from the F-layer are modeled separately and subtracted from the measurements. To account for residuals that may remain after subtraction, the DIPCont package contains a parameter $N_{eF}$.

## 2.5 Ion temperature

Temperature profiles obtained by the International Reference Ionosphere (IRI) 2.0 model (Bilitza et al., 2022) indicate that ion and neutral temperatures are very similar throughout the LTI, see Figures S2a–S2d in the supplementary material to this report. In analogy with the neutral temperature case, ion temperature $T_i$ is assumed to vary linearly with altitude $z$:

$$T_i(z|z_0, T_{i0}, L_{i0}) = T_{i0} \cdot \left(1 + \frac{z - z_0}{L_{i0}}\right). \tag{16}$$

The parameters $T_{i0}$ and $L_{i0}$ are the ion temperature and the gradient length scale, respectively, at the chosen reference altitude $z_0$.

## 2.6 Ion-neutral collision frequency

In quantitative terms, collision processes in the partially ionized LTI medium remain inadequately described, and are major sources of uncertainties in empirical models (e.g., Palmroth et al., 2021; Heelis and Maute, 2020; Sarris, 2019). At this stage, the DIPCont project is less concerned with optimizing the quantitative description of the LTI, but rather with the quality of parameter estimation extrapolation. While the choice of the best LTI model is certainly important for recovering the real values of targeted observables, further work will be needed, by parametric studies, comparison with previous work, and data analysis when a low-perigee mission such as Daedalus (Sarris et al., 2020) provides in situ measurements in the LTI. For our goal here, the chosen variant among the models for ion-neutral collision frequency $\nu_{in}$ should not matter too much as long as the underlying variability associated with erroneous measurements is captured. To this end, we follow the description of Huba (2019) and write

$$\nu_{in} = \sigma_{in} N_n \sqrt{\frac{kT_i}{m_i}} \tag{17}$$

with the collision cross section $\sigma_{in} \sim 5 \cdot 10^{-15}\,\mathrm{cm}^2$. An even simpler expression could neglect the variation with ion temperature $T_i$ so that $\nu_{in}$ becomes directly proportional to the neutral density $N_n$.

## 2.7 Pedersen conductivity

Using the approximations explained at the beginning of Section 2, Pedersen conductivity is given by

$$\sigma_{\mathrm{P}} = \frac{N_e e^2}{m_i} \frac{\nu_{in}}{\nu_{in}^2 + \Omega_i^2} \tag{18}$$

for a quasi-neutral two-component plasma when the contribution from electron-neutral collisions is neglected, see also Eq. (1), reproduced here for convenience. Compared to other variables and parameters of the LTI models presented here, the dependence of ion gyrofrequency $\Omega_i = q_i B / m_i$ ($q_i$: ion charge, $m_i$: ion mass) on magnetic field strength $B$ can be determined from

| Symbol | Description |
|---|---|
| $z_0$ | Local reference altitude |
| $T_{n0}$ | Neutral temperature at $z_0$ |
| $L_{n0}$ | Neutral temperature gradient length at $z_0$ |
| $N_{n0}$ | Neutral density at $z_0$ |
| $H_{n0}^N$ | Neutral density scale height at $z_0$ |
| $\eta = \eta_n$ | Inverse gradient of neutral density scale height |
| $N_{e0}$ | Electron density at $z_0$ |
| $L_{r0}$ | Radiation absorption length at $z_0$ |
| $\chi$ | Inclination angle of incident radiation |
| $N_{eF}$ | F-layer contribution to electron density |
| $T_{i0}$ | Ion temperature at $z_0$ |
| $L_{i0}$ | Ion temperature gradient length at $z_0$ |

**Table 1.** Parameters of model functions in local representation. The list is partially redundant, e.g., $L_{n0} = \eta H_{n0}^N$. The parameters $L_{r0}$ and $\chi$ cannot be estimated independently but only in combination $L_{r0} \cos \chi$. Boundary data (neutral and ion temperatures at the base of the LTI) are used to constrain the parameters $L_{n0}$, $\eta$, $H_{n0}^N$, and $L_{i0}$, see Section 3.3.

measurements or models of the magnetic field with very good accuracy, hence the associated variability should not much affect our results. Furthermore, in the logic of the LTI model constructed for the initial version of the DIPCont package, changes
in atmospheric composition and thus average ion mass are disregarded. Inspection of Figures S2a–S2d in the supplementary material to this report indicate that in the lower part of the LTI (altitudes below about 150 km) being the focus of downward continuation quality in the current study, variations of average ion mass with altitude are relatively small. Hence, altitude variations of ion gyrofrequency are neglected. In the same way as for other LTI model variables, namely, through the dependence of the parameters in the vector $\mathbf{p} = \mathbf{p}(x)$ (see Subsection 2.8 below) on the coordinate $x$, horizontal variations of magnetic field
strength $B$ and thus ion gyrofrequency $\Omega_i$ can be modeled, and are planned to be considered in future work.

## 2.8    LTI model in compact form

Parameters of model functions in local representation are listed in Table 1.

     The description of the DIPCont modeling procedure in Section 3 benefits from summarizing the LTI model in compact form as $\mathbf{m} = \mathbf{m}(z|\mathbf{p})$, with parameters $T_{n0}, H_{n0}^N, \eta, \ldots$, entering the vector $\mathbf{p}$. The parametric functions $T_n(z), N_n(z), N_e(z), T_i(z)$,
$\nu_{in}(z) = \nu_{in}(N_n(z), T_i(z))$, and $\sigma_P(z) = \sigma_P(N_e(z), \nu_{in}(z))$ constitute the components of the vectorial function $\mathbf{m}$.

## 3 DIPCont modeling procedure

The DIPCont modeling procedure is as follows.

– Synthetic noise-free measurements $\mathbf{m}_j = \mathbf{m}(z_j|\mathbf{p})$ are created along anticipated Daedalus satellite orbit sections around perigee at altitudes $z_j = z(t_j)$ and horizontal distances $x_j = x(t_j)$. The chosen model parameters are defined by vectors $\mathbf{p} = \mathbf{p}(x_\#)$ on a grid of horizontal distances $x_\#$. The integration and approximation methods employed for constructing the satellite orbits are described in Section 3.1 and in Appendix C.

– Using the multiplicative noise model presented in Section 3.2, synthetic measurements are contaminated by random errors in accordance with relative uncertainties specified in the Daedalus Report for Assessment (ESA, 2020), yielding ensembles $\{\tilde{\mathbf{m}}_j^k\}$ of noisy synthetic data sets.

– For a point $x_\#$ on the horizontal grid, synthetic data with horizontal distances $x_j$ in $[x_\# - \Delta x, x_\# + \Delta x]$ are considered to produce a least-squares estimate $\hat{\mathbf{p}}^k(x_\#)$ of the parameter vector $\mathbf{p}(x_\#)$. Repeating the estimation procedure for all members $k$ of the ensemble $\{\tilde{\mathbf{m}}_j^k\}$ of synthetic data sets yields ensembles of model parameters $\{\hat{\mathbf{p}}^k(x_\#)\}$ for all horizontal grid points $x_\#$. Specifics of the estimation procedure are discussed in Section 3.3.

– With parameter vectors $\mathbf{p} \in \{\hat{\mathbf{p}}^k(x_\#)\}$, the parametric model function $\mathbf{m} = \mathbf{m}(z|\mathbf{p})$ can be evaluated to obtain ensembles $\{\hat{\mathbf{m}}^k(z, x_\#)\} = \{\mathbf{m}(z|\hat{\mathbf{p}}^k(x_\#))\}$, representing altitude profiles of LTI observables and derived variables such as $\nu_{in}$ and $\sigma_\mathrm{P}$ over the entire range of LTI altitudes, and for all horizontal grid points $x_\#$. The resulting altitude profiles form a representative ensemble in the sense that their statistics are compatible with the model functions and the set of given relative errors. Relative deviation measures of observables and derived variables as functions of altitude are constructed. Finally, the concept of extrapolation horizons, introduced in Section 3.4, captures the altitude range where errors are tolerable according to predefined thresholds.

### 3.1 Satellite orbits around perigee

The DIPCont model offers two options for computing altitudes and horizontal distances along the orbits of satellites around perigee, namely, numerical integration by means of the Störmer-Verlet method (e.g., Hairer et al., 2003), and the polynomial approximation

$$z(t) = z_\mathrm{per} + \frac{a_\mathrm{per}}{2} t^2 \,, \tag{19}$$

$$x(t) = \frac{R_\mathrm{E} V_\mathrm{per} t}{R_\mathrm{per}} \left(1 - \frac{a_\mathrm{per}}{3 R_\mathrm{per}} t^2\right) \,, \tag{20}$$

with the acceleration $a_\mathrm{per}$ at perigee given by

$$a_\mathrm{per} = \frac{GM_\mathrm{E}}{R_\mathrm{per}^2} \frac{R_\mathrm{apo} - R_\mathrm{per}}{R_\mathrm{apo} + R_\mathrm{per}} = g_\mathrm{per} \varepsilon \,, \tag{21}$$

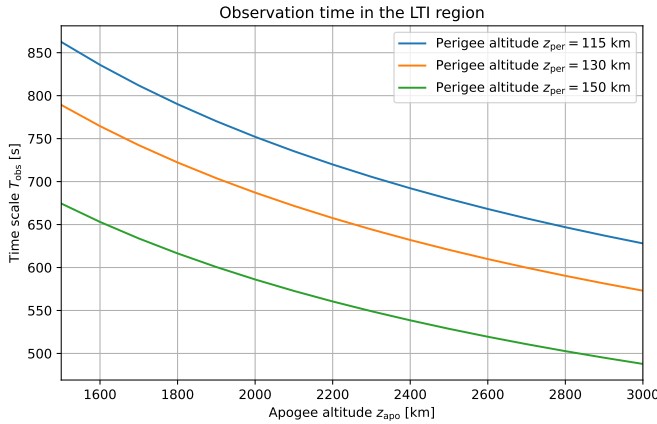

**Figure 3.** Observation time $T_{\text{obs}}$ in the LTI versus apogee altitude $z_{\text{apo}}$ for three different values of perigee altitude $z_{\text{per}}$. The topside of the LTI is assumed to be at $z_{\text{T}} = 200\,\text{km}$.

see Appendix C. Here $R_{\text{E}}$ is the Earth's radius, $M_{\text{E}}$ is the Earth's mass, $G$ is the gravitational constant, $z_{\text{per}}$, $R_{\text{per}}$, and $V_{\text{per}}$ are the altitude, geocentric distance, satellite velocity at perigee, $g_{\text{per}} = \frac{GM_{\text{E}}}{R_{\text{per}}^2}$ is the Earth's gravitational acceleration at geocentric distance $R_{\text{per}}$, $R_{\text{apo}}$ is the geocentric distance at apogee, and $\varepsilon = \frac{R_{\text{apo}} - R_{\text{per}}}{R_{\text{apo}} + R_{\text{per}}}$ is the orbital eccentricity. For the parameter range considered in this study, the deviation of the polynomial approximation from the more precise orbit integration is on the order of a few hundred meters, see Figure S4b in the supplementary material to this report.

The observation time $T_{\text{obs}}$ spent by Daedalus in the LTI during a perigee pass controls the amount of data that can be gathered for statistical investigations. Using the quadratic orbital approximation around perigee, $T_{\text{obs}}$ is twice the time needed to move from $z = z_{\text{per}}$ to the upper boundary at $z = z_{\text{T}}$, thus $z_{\text{T}} - z_{\text{per}} = \frac{a_{\text{per}}}{2}(T_{\text{obs}}/2)^2$, $T_{\text{obs}}^2 = \frac{8(z_{\text{T}} - z_{\text{per}})}{a_{\text{per}}}$, and

$$T_{\text{obs}}^2 = \frac{8(z_{\text{T}} - z_{\text{per}})R_{\text{per}}^2}{GM_{\text{E}}}\frac{R_{\text{apo}} + R_{\text{per}}}{R_{\text{apo}} - R_{\text{per}}} = \frac{8(z_{\text{T}} - z_{\text{per}})}{g_{\text{per}}\,\varepsilon}\ . \tag{22}$$

The variations of $T_{\text{obs}}$ with apogee altitude in the range $1500\,\text{km} \leq z_{\text{apo}} \leq 3000\,\text{km}$ for the three perigee altitudes $z_{\text{per}} = 115, 130, 150\,\text{km}$ are displayed in Figure 3. Raising the perigee from $115\,\text{km}$ to $130\,\text{km}$ yields a small reduction of observation time by about 10%. Within the range of orbital parameters considered here, the overall amount of data gathered during a perigee pass turns out to depend only moderately on apogee altitude $z_{\text{apo}}$, with a relative difference of not more than about 20% for changes in $z_{\text{apo}}$ between $2000\,\text{km}$ and $3000\,\text{km}$.

When dual-satellite missions to the LTI are considered, the question arises how synchronous the measurements are with respect to ground horizontal distance $x$, assuming the two spacecraft share the same orbital plane, have identical semi-major axes and thus orbital periods, and pass through their perigees at the same time. Figure S4a in the supplementary material to this report illustrates how visit times of ground horizontal distances are expected to differ for two satellites with perigee altitudes $130\,\text{km}$ and $150\,\text{km}$. Differences of satellite visit times turn out to be on the order of seconds.

| Observable | Relative error |
|---|---|
| Neutral temperature $T_n$ | 0.2 |
| Neutral density $N_n$ | 0.2 |
| Electron density $N_e$ | 0.1 |
| Ion temperature $T_i$ | 0.1 |

**Table 2.** Relative error levels used in this study, according to Table 2 of the Daedalus Report for Assessment (ESA, 2020).

## 3.2 Synthetic measurements and positivity constraints

Synthetic measurements $\{\tilde{\mu}_1, \tilde{\mu}_2, \tilde{\mu}_3, \ldots\}$ of an observable at altitudes $\{z_1, z_2, z_3, \ldots\}$ are constructed from a parametric model function $\mu = \mu(z|\mathbf{p})$ producing predictions that are contaminated by random errors $\{\sigma_1, \sigma_2, \sigma_3, \ldots\}$ from a suitable probability distribution. The model parameter vector $\mathbf{p}$ is estimated through minimization of a cost function. Following the standard least squares approach, the cost function is chosen to be the error-scaled square deviation

$$\chi^2(\mathbf{p}) \;=\; \sum_j \left( \frac{\tilde{\mu}_j - \mu(z_j|\mathbf{p})}{\sigma_j} \right)^2 . \tag{23}$$

The observables of interest $T_n$, $N_n$, $N_e$, $T_i$ are all positive, hence a straightforward additive noise model would not be appropriate as it may produce negative synthetic data. Furthermore, instrumental uncertainties as provided in the Daedalus Report for Assessment (ESA, 2020) are typically specified as relative (multiplicative) errors. Both issues are addressed by considering as model predictions $\mu_j = \mu(z_j|\mathbf{p})$ and data $\tilde{\mu}_j$ not the positive observables as such but their (natural) logarithms, and relative uncertainties for the random errors $\{\sigma_1, \sigma_2, \sigma_3, \ldots\}$. In the case of a (neutral or electron) density $N$, one obtains

$$\ln \tilde{N}_j \;=\; \ln N(z_j) + \sigma_j r_j \tag{24}$$

where the $r_j \sim \mathcal{N}(0,1)$ represent Gaussian noise (normally distributed random numbers with zero mean and unit variance), and $N = N(z)$ refers to the (positive) density model. Then

$$\tilde{N}_j \;=\; e^{\sigma_j r_j} \cdot N(z_j) \tag{25}$$

so that positivity is guaranteed. Furthermore,

$$e^{\sigma_j r_j} \approx 1 + \sigma_j r_j \,, \tag{26}$$

showing that the parameters $\sigma_j$ correspond to relative error levels. Table 2 summarizes the values used in this report.

In general, the parameters enter the logarithms of model functions nonlinearly, and an iterative estimation procedure is required.

### 3.3 Parameter estimation strategies

The model parameters listed in Table 1 are estimated from observations of neutral temperature $T_n$, neutral density $N_n$, electron density $N_e$, and ion temperature $T_i$ as follows.

- For a given horizontal grid location $x_\#$, data within the interval $[x_\# - \Delta x, x_\# + \Delta x]$ are considered. The effective window width is $2\Delta x$, see the white solid rectangles in Figures 1 and 4.

- From $T_n$ data and constraining the neutral temperature profile at the LTI lower boundary $z_B$ as explained below, infer $T_{n0}$, $H_{n0}^N$ and $\eta$. See Eq. (10) and Section 2.1.

- Using $H_{n0}^N$ and $\eta$, estimate $N_{n0}$ from $N_n$ data. See Eq. (13).

- Using $H_{n0}^N$ and $\eta$, estimate $N_{e0}$ and $L_{r0} \cos\chi$ from $N_e$ data. See Eq. (14).

- From $T_i$ data and a suitable constraint at the LTI lower boundary $z_B$ in analogy to the neutral temperature case, infer $T_{i0}$ and $L_{i0}$. See Eq. (16).

Altitude profiles of these observables allow for constructing the height dependence of derived variables such as ion-neutral collision frequency $\nu_{in}$ and the Pedersen conductivity $\sigma_P$, see Eqs. (17) and (18), respectively.

**Lower LTI boundary constraints**

As explained in Appendix A, Eqs. (A10) and (A13), the linear density scale height profile can be parametrized using $H_{n0}^N$ and $\eta$ in the form

$$H_n^N(z|z_0, H_{n0}^N, \eta) \;=\; H_{n0}^N \cdot \left(1 + \frac{z - z_0}{\eta H_{n0}^N}\right) \; . \tag{27}$$

It is important to note that the $H_n^N$ profile takes center stage in the LTI models of the observables $T_n$, $N_n$, and $N_e$. While the local temperature amplitude $T_{n0}$ is essentially an average of local temperature data around an altitude $z_0$, and the same applies to the local pressure scale height $H_{n0}^P$ obtained from $T_{n0}$ by simple multiplication, the inverse density scale height gradient $\eta$ and thus also the local density scale height parameter $H_{n0}^N = \frac{\eta-1}{\eta} H_{n0}^P$ are very challenging to estimate from purely local data with little variance in altitude, as suggested already by the standard error of the slope in linear regression analysis. Fortunately, neutral temperature at the base of the LTI is known with reasonable tolerances from atmospheric models (e.g., Picone et al., 2002; Emmert et al., 2021). This remote data point constitutes a valuable constraint for estimating the density scale height profile. To incorporate model uncertainties and expected deviations from actual values, boundary data at the base of the LTI are contaminated by random errors according to the approach described in Section 3.2.

To be specific, the pressure scale height gradient, constant under the assumptions discussed in Section 2.1 and Appendix A, can be obtained from its values $H_{n0}^P$ and $H_{nB}^P$ at $z_0$ and the LTI base altitude $z_B$, respectively, as follows:

$$\frac{\mathrm{d}H_n^P}{\mathrm{d}z} \;=\; \frac{H_{n0}^P - H_{nB}^P}{z_0 - z_B} \; . \tag{28}$$

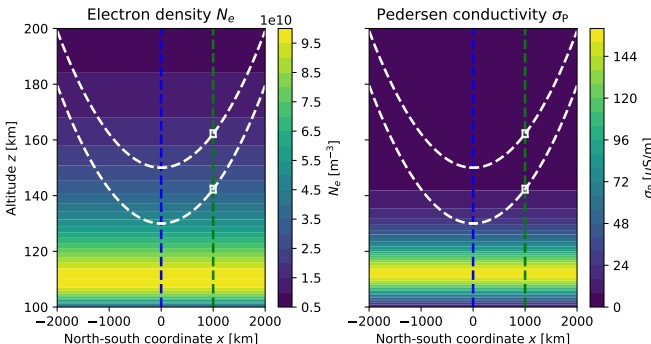

**Figure 4.** Model distributions of electron density $N_e$ (left panel) and Pedersen conducitivity $\sigma_P$ (right panel) in the LTI. Synthetic measurements are produced along the two satellite orbits (white dashed lines). The parameters of vertical profiles are estimated using measurements within a window (white solid rectangle) around two locations in horizontal direction (blue and green dashed lines).

The inverse gradients $\gamma$ and $\eta$ of pressure scale height and density scale height, respectively, are related by Eq. (8) through $\eta = \gamma + 1$, thus the parameter $\eta$ is given by

$$\eta \;=\; \frac{z_0 - z_{\mathrm{B}}}{H_{n0}^P - H_{n\mathrm{B}}^P} + 1 \;=\; \frac{M_n g}{R_{\mathrm{gas}}}\frac{z_0 - z_{\mathrm{B}}}{T_{n0} - T_{n\mathrm{B}}} + 1 \;, \tag{29}$$

where $T_{n\mathrm{B}}$ denotes the neutral temperature at $z_{\mathrm{B}}$. The local density scale height $H_{n0}^N$ can now be obtained from Eq. (9) as

$$H_{n0}^N \;=\; \frac{H_{n0}^P}{1 + \gamma^{-1}} \;=\; \frac{\eta - 1}{\eta}\frac{R_{\mathrm{gas}} T_{n0}}{M_n g} \;. \tag{30}$$

**Linear estimation of electron density parameters**

The logarithm of the electron density model considered here,

$$\ln N_e(z) \;=\; \ln N_{e0} + \frac{1}{2}\frac{\eta}{\eta - 1}\left[-\theta_0 + \frac{H_{n0}^N}{L_{r0}\cos\chi}\left(1 - e^{-\theta_0}\right)\right] \;, \tag{31}$$

can be combined with the logarithm of the neutral density model,

$$\ln N_n(z) \;=\; \ln N_{n0} - \frac{\eta}{\eta - 1}\theta_0 \;, \tag{32}$$

to find

$$\begin{aligned}
&\ln N_e(z) - \tfrac{1}{2}\ln N_n(z) \\
&= \ln N_{e0} - \tfrac{1}{2}\ln N_{n0} + \tfrac{1}{2}\tfrac{\eta}{\eta - 1}\tfrac{H_{n0}^N}{L_{r0}\cos\chi}\left(1 - e^{-\theta_0}\right) \\
&= a + b\left(1 - e^{-\theta_0}\right) \;,
\end{aligned} \tag{33}$$

showing that $a = \ln N_{e0} - \tfrac{1}{2}\ln N_{n0}$ and $b = \tfrac{1}{2}\tfrac{\eta}{\eta - 1}\tfrac{H_{n0}^N}{L_{r0}\cos\chi}$ can be obtained from linear regression of $\ln N_e - \tfrac{1}{2}\ln N_n$ versus $1 -$
$e^{-\theta_0}$ with $\theta_0 = \theta_0(z) = (\eta - 1)\ln\left(1 + \frac{z - z_0}{\eta H_{n0}^N}\right)$. Since the parameters $\eta$ and $H_{n0}^N$ are available as estimates from $T_n$ modeling,

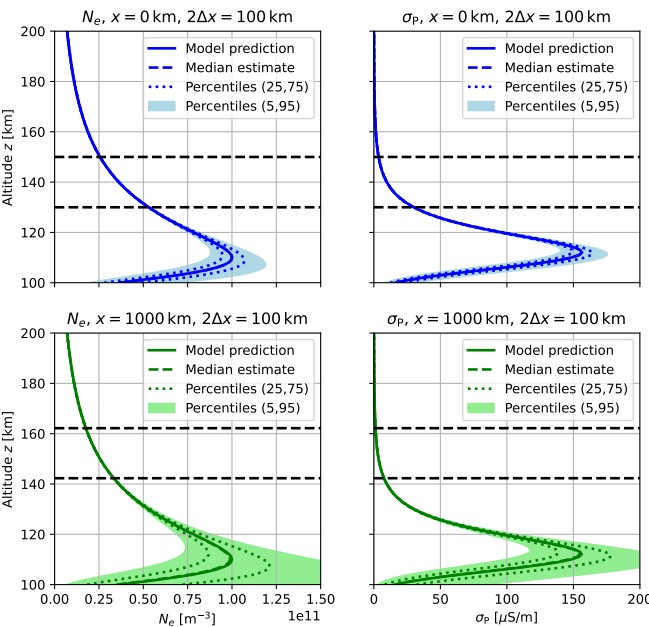

**Figure 5.** Visualization of the ensemble of altitude profiles generated from the Monte Carlo distributions of model parameters. Shown are selected quantiles evaluated at the vertical grid of LTI altitudes. Left panels: electron density $N_e$. Right panels: Pedersen conductivity $\sigma_\mathrm{P}$. Upper panels: center position (blue dashed line) in Figure 4. Lower panels: right position (green dashed line) in Figure 4.

and $N_{n0}$ is known from $N_n$ modeling, $N_{e0}$ and $L_{r0}\cos\chi$ can be computed from the linear coefficients $a$ and $b$, hence this special case does not necessitate an iterative parameter estimation approach.

### 3.4 Error profiles and extrapolation horizons

With $\{\tilde{\mathbf{m}}_j^k\}_{j\in[\#]}^{k=1}$ being a single set ($k=1$) of synthetic measurements, and $j\in[\#]$ indicating that horizontal distances are selected to be within $\pm\Delta x$ around a predefined grid point $x_\#$, the estimation procedure yields a specific estimate $\hat{\mathbf{p}}^k$ of the parameter vector $\mathbf{p}(x_\#)$. In a Monte Carlo setup, different instances of random errors are applied to the model predictions to produce data sets $\{\tilde{\mathbf{m}}_j^1,\tilde{\mathbf{m}}_j^2,\tilde{\mathbf{m}}_j^3,\dots\}_{j\in[\#]}$. The ensemble of data sets gives rise to an ensemble of parameter vectors $\{\hat{\mathbf{p}}^k\}=\{\hat{\mathbf{p}}^1,\hat{\mathbf{p}}^2,\hat{\mathbf{p}}^3,\dots\}$, which in turn, when entered in $\mathbf{m}=\mathbf{m}(z|\mathbf{p})$, yields an ensemble of profiles $\{\hat{\mathbf{m}}^k(z,x_\#)\}=\{\hat{\mathbf{m}}^1(z,x_\#),\hat{\mathbf{m}}^2(z,x_\#),\hat{\mathbf{m}}^3(z,x_\#),\dots\}$ for the entire range of altitudes $z$, and at each point $x_\#$ of the horizontal coordinate grid.

The procedure is illustrated in Figures 4 and 5. Figure 4 shows the model functions and the satellite orbits used for computing the predictions that enter the Monte Carlo simulation. The ensemble of altitude profiles generated from the Monte Carlo distributions of model parameters is visualized in Figure 5 by means of selected quantiles evaluated at the vertical grid of LTI altitudes.

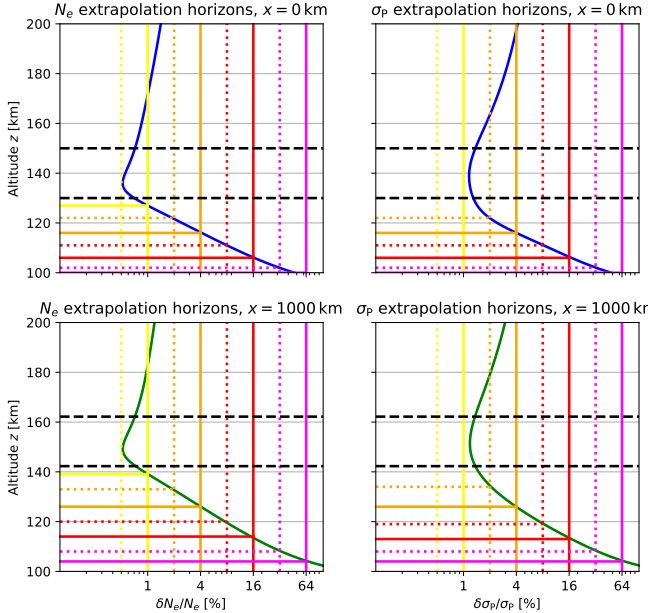

**Figure 6.** Solid lines (blue and green) give the relative root-mean-square deviations of Monte Carlo altitude profiles from the respective input model profiles at two horizontal locations. Vertical dotted and solid lines represent a set of chosen error levels, ranging from 0.5 % and 1 % (yellow) to 32% and 64% (magenta). The corresponding horizontal lines show the extrapolation horizons indicating at which altitude the relative deviation equals the respective error level. Left panels: electron density $N_e$. Right panels: Pedersen conducitivity $\sigma_P$. Upper panels: center position (blue dashed line) in Figure 4. Lower panels: right position (green dashed line) in Figure 4.

The ensemble of altitude profiles forms the basis for quantifying extrapolation quality through measures of relative deviation from a model prediction. Suppressing altitude and horizontal grid dependencies, and considering only a single model variable $\mu$ with ensemble members $\hat{\mu}^1, \hat{\mu}^2, \hat{\mu}^3, \ldots, \hat{\mu}^K$, the root-mean-square deviation is given by

$$\delta\mu = \sqrt{\langle(\hat{\mu}-\mu)^2\rangle} = \sqrt{\frac{1}{K}\sum_{k=1}^{K}(\hat{\mu}^k-\mu)^2} \,. \tag{34}$$

Figure 6 shows the altitude profiles of *relative* root-mean-square deviation $\delta\mu/\mu = \sqrt{\langle(\hat{\mu}-\mu)^2\rangle}/\mu$ for the variables and horizontal locations as in Figures 4 and 5.

Figure S3 in the supplementary material to this report provides additional information on this DIPCont model run, visualizing model distributions, ensembles of altitude profiles, and extrapolation horizons also for neutral temperature $T_n$, neutral density $N_n$, ion temperature $T_i$, and ion-neutral collision frequency $\nu_{in}$.

Alternative relative deviation measures considered in the DIPCont package are based on the empirical distribution of absolute deviations $|\hat{\mu} - \mu|$, e.g., the average absolute deviation from the model prediction $\mu$:

$$(\delta\mu)_{\text{abs}} \;=\; \langle|\hat{\mu} - \mu|\rangle \;=\; \frac{1}{K}\sum_{k=1}^{K}|\hat{\mu}^k - \mu|\,, \tag{35}$$

or selected quantiles of the distribution.

### 3.5 Implementation

The DIPCont model is implemented as a bundle of Python instuctions and functions collected in three modules.

In the module `DIPContBas.py`, the basic setup of the DIPCont framework is defined, e.g., LTI region boundaries and boundary values, satellite orbit parameters, horizontal grid locations, and auxiliary plot parameters. Furthermore, it also provides configurational variables that are exchanged between DIPCont functions and modules, e.g., parameters shared by different parametric models.

The module `DIPContMod.py` provides parametric model functions of LTI variables and plot routines.

The module `DIPContEst.py` is concerned with Monte Carlo parameter estimation and profile continuation. Estimation of parameters that enter the model functions nonlinearly is accomplished by the function `curve_fit()` from the module `scipy.optimize` whereas linear parameter estimation is performed using the function `linregress` from the module `scipy.stats`. Monte Carlo ensembles of parameters and altitude profiles are stored in `pandas` dataframes.

The three DIPCont modules are provided as supplementary files to this report, together with Jupyter notebooks to explain and illustrate their usage.

## 4   First results

The major ingredients of the DIPCont processing chain, namely, generation of synthetic in situ measurements along satellite orbits, Monte Carlo simulations of vertical profiles, and construction of extrapolation horizons, are summarized in Figures 4–6 displaying electron density $N_e$ and Pedersen conductivity $\sigma_{\text{P}}$ as two variables of key importance for the structure and the dynamics of the LTI. As indicated by Eq. (1) and the respective profiles in Figure 5, electron density makes the main contribution to the peaked height variation of Pedersen conductivity, with secondary contributions of neutral density and possibly ion temperature through the parametric form chosen for the ion-neutral collision frequency, see Section 2.6, and also Figure S3 in the supplementary material to this report. Furthermore, Pedersen conductivity controls the height variation of Joule heating, whose characterization is one of the main scientific targets of the proposed Daedalus mission (ESA, 2020). In the neutral wind reference frame, Joule heating is $\mathbf{j}_\perp \cdot \mathbf{E}_\perp = \sigma_{\text{P}}|\mathbf{E}_\perp|^2$ where the subscript $\perp$ indicates a vectorial component perpendicular to the ambient magnetic field direction $\hat{\mathbf{B}}$. Height variations of $\mathbf{E}_\perp$ are negligible according to the following rationale, see, e.g., Rishbeth (1997). Due to high parallel conductivity, the electric field component $E_\parallel = E_s$ parallel to $\hat{\mathbf{B}}$ vanishes, i.e., $0 = E_s = -\frac{\partial\Phi}{\partial s}$, where $s$ is the magnetic field line coordinate, and $\Phi$ denotes the electric potential. The electric field component $E_q$ in a direction perpendicular to $\hat{\mathbf{B}}$ captured by a coordinate $q$ then satisfies $\frac{\partial E_q}{\partial s} = -\frac{\partial}{\partial s}\frac{\partial\Phi}{\partial q} = -\frac{\partial}{\partial q}\frac{\partial\Phi}{\partial s} = \frac{\partial E_s}{\partial q} = 0$.

When instead of two selected horizontal locations as in Figure 6 an equidistant grid of horizontal coordinates is defined for DIPCont simulations and the construction of extrapolations horizons, the results can be displayed together with the underlying model distributions and satellite orbits as in Figure 1. In the following examples, such displays are used to visualize DIPCont results for different spacecraft configurations. Section 4.1 offers a first qualititative assessment of extrapolation quality in terms of varying inter-spacecraft distance. Section 4.2 contrasts the performance of the dual-spacecraft configuration considered so

far with the results of the single-spacecraft case.

       Note that the horizontal axis corresponds to the latitudinal (north-south) direction. In the simulations that led to Figures 4–6, horizontal variations were disregarded for better comparability. In Figure 1 and in the following, latitudinal inhomogeneity of electron density is meant to reproduce the two maxima observed by a polar orbiting satellite when crossing the auroral oval. The highest latitude corresponds to the origin of the horizontal axis. Since the physics of energetic particle precipitation is not incorporated in this initial version of the DIPCont package, the horizontal variation of electron density expected for

an auroral oval crossing is prescribed through ad hoc choices of horizontal electron density peak parameters profiles, see the option `LTIModelType='NeAuroralZoneCrossing'` in the DIPCont code as part of the supplementary material to this report. The functional forms of horizontal electron density peak parameters are given in Appendix D.

## 4.1   Varying inter-spacecraft distance

Extrapolation of two-point measurements is expected to perform best if the spatial separation matches the relevant physical length scale. In the LTI this should be the (local) density scale height, in the range of 10–20 km for altitudes above 130 km, as in our example of a dual-spacecraft setup with perigee altitudes of 130 km and 150 km, see Figure 1. The inter-spacecraft distance remains close to 20 km throughout the whole orbit section and thus also to the density scale height as the relevant physical scale. Note that in all dual-satellite DIPCont model runs presented in this paper, apogee distances of the second satellite have

been adjusted such that the sum of perigee and apogee distances are identical for both satellites, and thus also the semi-major axes and the orbital periods.

       Figure 7 displays extrapolation horizons for the same simulation setup except that the perigee altitude of the second satellite is reduced to 135 km, producing an inter-spacecraft distance at perigee of only 5 km. The separation is now smaller than the local density scale height with values of about 15 km at altitudes around 150 km. Compared to Figure 1, the errors are increased

and the extrapolation horizons reduced. The changes are not dramatic but enough to show that inter-spacecraft distance is a parameter to be considered when extrapolation quality is supposed to be optimized.

## 4.2   Single-satellite case

       To check how much a second satellite improves extrapolation quality, the Monte Carlo simulations summarized in Figure 1 are repeated for the single-spacecraft case, with all other parameters left unchanged. The resulting extrapolation horizons are

shown in Figure 8. Compared to ionospheric profile continuation from dual-spacecraft observations, the single-spacecraft case yields significantly worse results, with extrapolation horizons collapsing into the orbit near the perigee due to lacking variability in altitudes. Away from the perigee, the orbital motion of the satellite during the time corresponding to the horizontal window

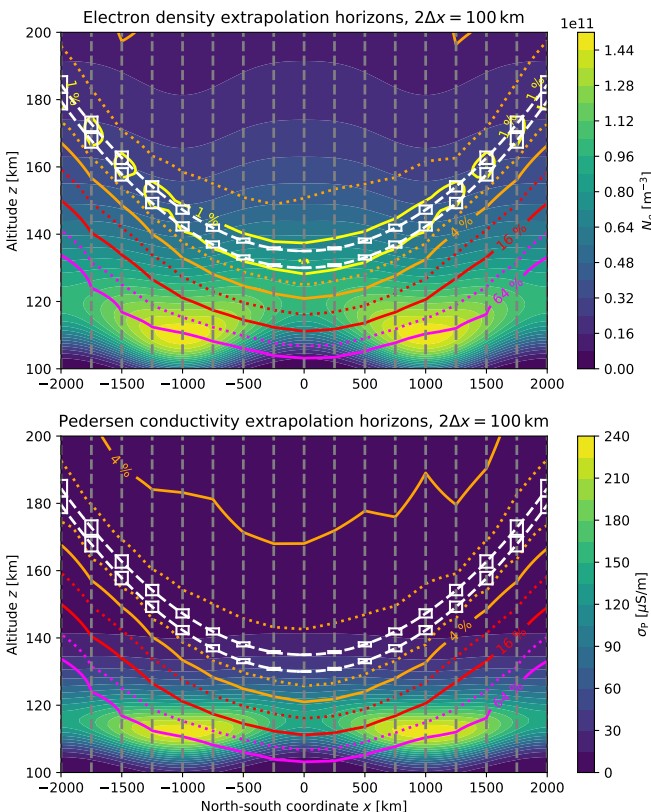

**Figure 7.** Same as Figure 1 but for an inter-spacecraft separation of 5 km at perigee.

width $2\Delta x$ yields some height range that allows for profile reconstruction but with significant errors. The peaks in electron density and Pedersen conductivity are clearly outside the largest considered error level of 64%, while Figure 1 shows that in
the dual-spacecraft case the peaks are between the 16% and 32% error levels.

## 5   Discussion

Our first results suggest that altitude profiles of key LTI variables can be reconstructed with sufficient accuracy from in situ measurements if the effective altitude range covers relevant physical scales such as the local density scale height $H_{n0}^{N}$. This is the case for a dual-spacecraft configuration with an inter-spacecraft separation of 20 km at perigee, see Figures 1 and 4–6.
By two-point sampling, one can retrieve the vertical profiles of electron density and Pedersen conductivity essentially down to the bottom of the LTI region, a few scale heights under the lower satellite and including the peak altitudes. For Pedersen conductivity, errors are expected in the range of several 10%, with the peak altitude and most of the conductivity within the 32% extrapolation horizon in the chosen example, consistent with rocket observations (Sangalli et al., 2009).

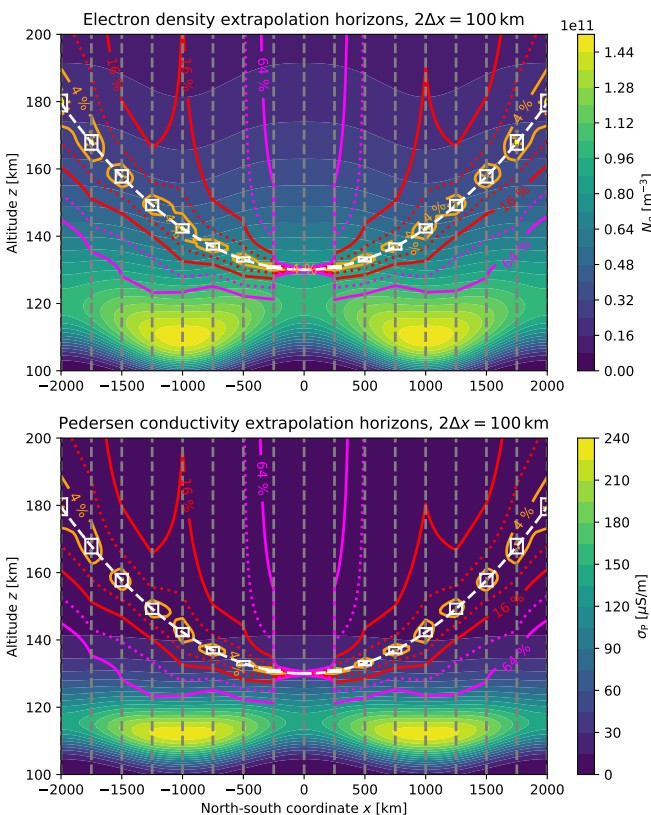

**Figure 8.** Same as Figure 1 but for the single-satellite case.

Given the current knowledge of key LTI variables, error levels of a few ten percent may well improve the situation. An important motivation behind the Daedalus proposal was the large error margin in Joule heating estimates, with a major contribution by errors in conductance (height-integrated conductivity). Thus, Sarris et al. (2020) pointed out that for a substorm event investigated by Palmroth et al. (2005), there were differences of up to 500% between three proxies of the Joule heating rates integrated over the Northern hemisphere. Even if this setup cannot be directly compared to our virtual environment, the order of magnitude difference between the two error margins looks encouraging for follow-up work on ionospheric profile continuation.

The DIPCont framework allows for addressing economical and technical questions regarding the impact of different LTI mission cost factors. On the one hand, a dual-spacecraft mission seems to automatically imply higher costs because a second satellite needs to be built. On the other hand, a major cost driver of any deep LTI mission is the necessary amount of propellant that is required in order to maintain a spacecraft in orbit, due to enhanced atmospheric drag at very low perigee altitudes. Since in a dual spacecraft setup the role of the lower perigee satellite can be shared, each of the two probes would have to carry half of the total amount of propellant required to maintain the same total observation time required by a single-satellite mission.

Moreover, the necessary amount of thermal shielding depends as well on perigee altitude and each of the two probes would have to withstand the maximum thermal stress at perigee less often. Our findings show that the two-point setup allows for a more effective extrapolation to lower altitudes, which in turn means that a higher perigee may well be a meaningful option.

Data processing would also benefit from raising the perigee. As shown by simulations carried out for the technical assessment of Daedalus (ESA, 2020), a hydrodynamic shock develops in front of the spacecraft at altitudes under $\sim$120–130 km, complicating the retrieval of unperturbed data from the observed ones. Another LTI mission parameter considered in this paper is the apogee altitude controlling the proximity to the Van Allen belts and thus the necessary amount of radiation shielding, but affecting also the available LTI observation time near perigee. The analysis presented in Section 3.1 shows that the amount of
data gathered for statistical studies depends only moderately on apogee altitude.

The current version of the DIPCont framework concentrates on the E-layer, assuming that contributions from the F-layer can be disregarded or subtracted before processing, e.g., using the NeQuick approach to model topside ionospheric sounding data (Pignalberi et al., 2020). The DIPCont package contains a parameter $N_{eF}$ to study the effect of F-layer residuals on extrapolation quality in future work.

The first results presented here are planned to be validated and extended in more extensive studies. Besides varying orbital parameters such as perigee altitude and inter-spacecraft distance, the impact of numerical parameters such as the horizontal selection window, $2\Delta x$, needs further investigation. As already commented in Section 2.6, alternative functional forms for modeling ion-neutral collision frequencies or other variables may also be considered.

## 6   Conclusions and Outlook

The DIPCont methodology introduced in this paper is designed to assess the quality of downward continuation of LTI variables using in situ satellite measurements and parametric models. While first results have been obtained with a simplified LTI description based on a single particle species, the Monte Carlo simulation machinery in DIPCont is not constrained to a particular model setup. By quantifying the quality of extrapolated in situ measurements, DIPCont can help to assess the science return of specific configurations and thus to optimize the parameters of upcoming LTI missions.

First DIPCont tests, performed on electron density and Pedersen conductivity, show promising results, to be consolidated by further parametric studies. Application of DIPCont to a modeled event, like the geomagnetic storm event of March 2015 addressed in the Daedalus Report for Assessment (ESA, 2020) is an upcoming target. This could be performed using the capabilities of the Daedalus MASE toolset (Sarris et al., 2023b). Future studies are planned to include Joule heating which was a major driver of the Daedalus mission proposal. To investigate auroral processes and the electrodynamics of magnetosphere-
ionosphere coupling, ionization through energetic particle precipitation needs to be incorporated. The Hall current nature of auroral electrojets calls for including electron-neutral collisional interaction as a major contributor to Hall conductivity formation.

Coordination between an LTI mission, like Daedalus, and a topside mission, e.g., like Swarm or DMSP, would enhance the return of both missions. As an example, reconstruction of vertical profiles of ionospheric conductivity based on LTI observa-

480 tions could help to calibrate topside estimates of the conductance, while topside electron density could provide upper continuation and constrain the height-integrated total electron content (TEC) inferred from LTI data. Combination with ground-based observatory data such as ionosondes would offer further valuable constraints to DIPCont, and thus enable more comprehensive modeling of the LTI.

*Code availability.* The DIPCont framework is implemented in three Python modules `DIPContBas.py`, `DIPContMod.py`, and `DIPContEst.py`.
The modules are provided as supplementary files to this report, together with Jupyter notebooks to explain and illustrate their usage. The DIPCont code is planned to be migrated to a public repository.

## Appendix A:  Neutral density profile for linear variations of scale height

Consider an atmospheric layer dominated by possibly several neutral constituents with an average or representative particle mass $m_n$, total pressure $P_n$, mass density $\varrho_n$, effective neutral number density $N_n = \varrho_n/m_n$ and temperature $T_n$. Under
490 hydrostatic conditions, $\mathrm{d}P_n = -\varrho_n g\,\mathrm{d}z$, where $z$ is altitude and $g$ is gravity (gravitational acceleration) assumed to vary so little within the layer that it can be safely considered constant. Using the ideal gas law $P_n = N_n k T_n$ where $k$ denotes the Boltzmann constant, one obtains $\mathrm{d}P_n = -P_n \frac{m_n g}{k T_n}\,\mathrm{d}z = -P_n \frac{\mathrm{d}z}{H_n^P}$ with the pressure scale height

$$H_n^P \;=\; \frac{kT_n}{m_n g}\,, \tag{A1}$$

Rearranging $-\frac{\mathrm{d}z}{H_n^P} = \frac{\mathrm{d}P_n}{P_n} = \mathrm{d}\ln P_n$ and integrating leads to

$$495 \quad P_n(z) \;=\; P_{n0} \exp\left\{ -\int_{z_0}^{z} \frac{\mathrm{d}\tilde{z}}{H_n^P(\tilde{z})} \right\} \tag{A2}$$

where the altitude dependence of $H_n^P$ directly reflects the change of temperature $T_n$ with $z$.

Analogous differential and integral expressions for the neutral density, namely, $\mathrm{d}\ln N_n = -\frac{\mathrm{d}z}{H_n^N}$ and

$$N_n(z) \;=\; N_{n0} \exp\left\{ -\int_{z_0}^{z} \frac{\mathrm{d}\tilde{z}}{H_n^N(\tilde{z})} \right\} \,, \tag{A3}$$

are derived as follows. Combining the differential of the ideal gas law $\mathrm{d}P_n = N_n k\,\mathrm{d}T_n + k T_n\,\mathrm{d}N_n$ with the hydrostatic condi-
500 tion yields $-N_n m_n g\,\mathrm{d}z = N_n k\,\mathrm{d}T_n + k T_n\,\mathrm{d}N_n$ and thus $-\frac{m_n g}{k T_n}\,\mathrm{d}z - \frac{1}{T_n}\,\mathrm{d}T_n = \frac{1}{N_n}\,\mathrm{d}N_n = \mathrm{d}\ln N_n$. Since $\frac{\mathrm{d}T_n}{T_n} = \mathrm{d}\ln T_n = \mathrm{d}\ln H^P = \frac{\mathrm{d}H_n^P}{H_n^P}$, one obtains

$$\frac{\mathrm{d}\ln N_n}{\mathrm{d}z} \;=\; -\frac{1}{H_n^P}\left(1 + \frac{\mathrm{d}H_n^P}{\mathrm{d}z}\right)\,. \tag{A4}$$

Therefore, the density scale height $H_n^N$ in the expression $\mathrm{d}\ln N_n = -\frac{\mathrm{d}z}{H_n^N}$ is given by

$$H_n^N \;=\; H_n^P\left(1 + \frac{\mathrm{d}H_n^P}{\mathrm{d}z}\right)^{-1}\,. \tag{A5}$$

To be more specific, we suppose the neutral temperature $T_n$ varies linearly with altitude $z$,

$$T_n(z) = T_{n0} \cdot \left(1 + \frac{z - z_0}{L_{n0}}\right) . \tag{A6}$$

where $T_{n0}$ is the temperature at a reference altitude $z_0$, and $L_{n0} = \frac{T_{n0}}{\mathrm{d}T_n/\mathrm{d}z}$ denotes the local gradient length. Then

$$H_n^P(z) = H_{n0}^P \cdot \left(1 + \frac{z - z_0}{L_{n0}}\right) . \tag{A7}$$

with

$$H_{n0}^P = \frac{kT_{n0}}{mg} , \tag{A8}$$

so that the pressure scale height gradient $\frac{\mathrm{d}H_n^P}{\mathrm{d}z} = \frac{H_{n0}^P}{L_{n0}}$ is constant, and thus also the gradient of density scale height:

$$\frac{\mathrm{d}H_n^N}{\mathrm{d}z} = \frac{\mathrm{d}H_n^P}{\mathrm{d}z} \cdot \left(1 + \frac{\mathrm{d}H_n^P}{\mathrm{d}z}\right)^{-1} . \tag{A9}$$

The linear profile of density scale height is given by

$$H_n^N(z) = H_{n0}^N \cdot \left(1 + \frac{z - z_0}{L_{n0}}\right) , \tag{A10}$$

with

$$H_{n0}^N = \frac{H_{n0}^P}{1 + H_{n0}^P/L_{n0}} = \frac{H_{n0}^P}{1 + \gamma^{-1}} . \tag{A11}$$

Here

$$\gamma = \left(\frac{\mathrm{d}H_n^P}{\mathrm{d}z}\right)^{-1} \tag{A12}$$

denotes the inverse gradient of pressure scale height. The inverse gradient of density scale height

$$\eta = \left(\frac{\mathrm{d}H_n^N}{\mathrm{d}z}\right)^{-1} = \frac{L_{n0}}{H_{n0}^N} \tag{A13}$$

is related to $\gamma$ through $\eta = \gamma + 1$.

In the non-isothermal case $L_{n0} < \infty$, integrating $1/H_n^N$ gives the expression

$$
\begin{aligned}
\zeta_0 & = \int_{z_0}^{z} \frac{\mathrm{d}\tilde{z}}{H_n^N(\tilde{z})} = \eta \ln\left(1 + \frac{z - z_0}{L_{n0}}\right) \\
& = -\ln\left(1 + \frac{z - z_0}{\eta H_{n0}^N}\right)^{-\eta} ,
\end{aligned}
\tag{A14}
$$

Hence, the altitude profile of number density (A3) is given by

$$N_n(z) = N_{n0} \cdot e^{-\zeta_0} = N_{n0} \cdot \left(1 + \frac{z - z_0}{\eta H_{n0}^N}\right)^{-\eta} . \tag{A15}$$

In the isothermal limit, $\eta \to \infty$, $\ln\left(1 + \frac{z-z_0}{\eta H_{n0}^N}\right) \to \frac{z-z_0}{\eta H_{n0}^N}$, thus $\zeta_0 \to \frac{z-z_0}{H_{n0}^N}$, and $H_{n0}^N \to H_{n0}^P$ through Eq. (A11).

## Appendix B:  Electron density profile for linear variations of scale height

Following the approach first presented by Chapman (1931), the ionization rate per unit volume $q$ is expressed in terms of the intensity $I$ of ionizing radiation, the ionization efficiency $\kappa$, the angle $\chi$ of incident radiation with the atmospheric layer normal vector, the radiation absorption cross-section $\sigma_r$, and the neutral density $N_n$ as $q = \kappa \cos\chi \frac{\mathrm{d}I}{\mathrm{d}z}$. Here $z$ is altitude, and the $z$ axis is pointing upwards as before. The function $q = q(z)$ is also called production function. Although originally proposed for photoionization, the Chapman approach may be applied also to ionization by precipitation of energetic particles as in the auroral region, if model variables and coefficients are properly interpreted.

The intensity $I$ satisfies the differential equation

$$\mathrm{d}I \;=\; \sigma_r\, N_n\, I\, \frac{\mathrm{d}z}{\cos\chi} \tag{B1}$$

with the solution

$$I(z) \;=\; I_\infty \exp\left\{ \frac{\sigma_r}{\cos\chi} \int_{z_\infty}^{z} N(\tilde{z})\mathrm{d}\tilde{z} \right\} \tag{B2}$$

where $z_\infty$ and $I_\infty$ refer to an upper boundary sufficiently remote from the atmospheric layer.

Using $\mathrm{d}I = \sigma_r N_n I \frac{\mathrm{d}z}{\cos\chi}$, the production function $q$ can be rewritten as $q = \kappa\sigma_r N_n I$ and thus

$$q(z) \;=\; \kappa\,\sigma_r\, N_n(z)\, I_\infty \exp\left\{ \frac{\sigma_r}{\cos\chi} \int_{z_\infty}^{z} N(\tilde{z})\mathrm{d}\tilde{z} \right\}\;. \tag{B3}$$

The ionization peak altitude $z_*$ is obtained from the condition

$$0 = \left.\frac{\mathrm{d}\ln q}{\mathrm{d}z}\right|_{z=z_*} = \frac{N_n'(z_*)}{N_n(z_*)} + \frac{\sigma_r N_n(z_*)}{\cos\chi} \tag{B4}$$

where the prime denotes differentiation with respect to altitude $z$. Considering Eqs. (A3) and (A14) gives rise to $N_n(z) = N_{n0}e^{-\zeta_0}$, $\zeta_0' = 1/H_n^N$, and defining the *radiation absorption length* $L_r = L_r(z)$ by

$$L_r \;=\; \frac{1}{\sigma_r N_n}\;, \tag{B5}$$

the general ionization peak condition is conveniently expressed as

$$H_n^N(z_*) \;=\; L_r(z_*)\cos\chi\;. \tag{B6}$$

### B1   Local representation of electron density

Assuming the neutral temperature $T_n$ varies linearly with altitude $z$, the altitude dependence of electron density was modeled by Gledhill and Szendrei (1950). Since their formulation does not fit well with the DIPCont nomenclature used in the current

report, an independent and extended derivation is presented now. Using $T_n(z) = T_{n0} \cdot \left(1 + \frac{z - z_0}{L_{n0}}\right) = T_{n0} \cdot \left(1 + \frac{z - z_0}{\eta H_{n0}^N}\right)$, and $\eta < \infty$, the altitude profile of neutral number density can be written in the form

$$N_n(z) = N_{n0} \left(1 + \frac{z - z_0}{\eta H_{n0}^N}\right)^{-\eta} , \tag{B7}$$

see Appendix A and Eq. (A15). Integration gives

$$\int_{z_\infty}^z N_n(\tilde{z}) \, \mathrm{d}\tilde{z} = -N_{n0} \frac{\eta H_{n0}^N}{\eta - 1} \left[ \left(1 + \frac{\tilde{z} - z_0}{\eta H_{n0}^N}\right)^{-(\eta - 1)} \right]_{\tilde{z} = z_\infty}^{\tilde{z} = z} . \tag{B8}$$

In this LTI modeling context it is safe to assume that the regional temperature increase with altitude is moderate enough to ensure $H_{n0}^N < L_{n0}$, then $\eta > 1$. Furthermore, the altitude $z_\infty$ is chosen to be large enough for the contribution from the value at $\tilde{z} = z_\infty$ to be negligible. We obtain

$$\int_{z_\infty}^z N_n(\tilde{z}) \, \mathrm{d}\tilde{z} = -N_{n0} \frac{\eta H_{n0}^N}{\eta - 1} \left(1 + \frac{z - z_0}{\eta H_{n0}^N}\right)^{-(\eta - 1)} \tag{B9}$$

by using Eq. (A14). Defining

$$\theta_0 = \frac{\eta - 1}{\eta} \zeta_0 = (\eta - 1) \ln\left(1 + \frac{z - z_0}{\eta H_{n0}^N}\right) , \tag{B10}$$

the radiation intensity profile assumes the form

$$I(z) = I_\infty \exp\left\{ -\frac{\sigma_r N_{n0}}{\cos\chi} \frac{\eta H_{n0}^N}{\eta - 1} e^{-\theta_0} \right\} \tag{B11}$$

$$= I_\infty \exp\left\{ -\frac{\eta}{\eta - 1} \frac{H_{n0}^N}{L_{r0} \cos\chi} e^{-\theta_0} \right\} \tag{B12}$$

where $L_{r0} = L_r(z_0)$. The neutral density (A15) is rewritten as

$$N_n(z) = N_{n0} \exp\left\{ -\frac{\eta}{\eta - 1} \theta_0 \right\} , \tag{B13}$$

so that the production function (B3) assumes the form

$$q(z) = \frac{\kappa I_\infty}{L_{r0}} \exp\left\{ \frac{\eta}{\eta - 1} \left[ -\theta_0 - \frac{H_{n0}^N}{L_{r0} \cos\chi} e^{-\theta_0} \right] \right\} . \tag{B14}$$

In the isothermal limit, $\eta \to \infty$, $\frac{\eta}{\eta - 1} \to 1$, $\theta_0 \to \frac{z - z_0}{H_{n0}^N}$, and the isothermal Chapman production function (Chapman, 1931) is recovered.

In static equilibrium of photoionization and quadratic recombination, $q = \alpha N_e^2$ with the recombination coefficient $\alpha$, thus $N_e = \sqrt{q/\alpha}$. Using

$$N_{e0} = N_e(z_0) = \sqrt{\frac{\kappa I_\infty}{\alpha L_{r0}}} \exp\left\{ -\frac{1}{2} \frac{\eta}{\eta - 1} \frac{H_{n0}^N}{L_{r0} \cos\chi} \right\} , \tag{B15}$$

we obtain

$$N_e(z) = N_{e0} \exp\left\{ \frac{1}{2} \frac{\eta}{\eta - 1} \left[ -\theta_0 + \frac{H_{n0}^N}{L_{r0} \cos\chi} \left(1 - e^{-\theta_0}\right) \right] \right\} . \tag{B16}$$

## B2  Representation of electron density in terms of ionization peak parameters

A meaningful regional representation of the electron density can be constructed by means of the ionization peak parameters. For a given incident radiation angle $\chi$, the altitude $z_*$ of the electron density maximum can be expressed in local parameters as follows:

$$z_* = z_0 + \eta H_{n0}^N \left[ \Gamma^{1/(\eta-1)} - 1 \right] . \tag{B17}$$

where

$$\Gamma = \frac{H_{n0}^N}{L_{r0} \cos \chi} . \tag{B18}$$

The electron density peak value $N_{e*} = N_e(z = z_*)$ is

$$N_{e*} = N_{e0} \exp \left\{ \frac{1}{2} \frac{\eta}{\eta - 1} \left[ -\ln \Gamma + \Gamma - 1 \right] \right\} . \tag{B19}$$

With $z_*$ as the reference altitude, $z_0 = z_*$, we can take advantage of the condition (B6) $H_{n0}^N = H_{n*}^N = L_{r*} \cos \chi = L_{r0} \cos \chi$, thus

$$N_e(z) = N_{e*} \exp \left\{ \frac{1}{2} \frac{\eta}{\eta - 1} \left[ -\theta_* + 1 - e^{-\theta_*} \right] \right\} , \tag{B20}$$

where $\theta_* = \theta_*(z) = (\eta - 1) \ln \left( 1 + \frac{z - z_*}{\eta H_{n*}^N} \right)$, and $H_{n*}^N$ denotes the density scale height at $z = z_*$. This representation shows that $\chi$ is only an implicit parameter of the electron density model, and cannot be inferred from knowledge of the peak parameters.

## Appendix C:  Orbit approximation around perigee

Consider a Kepler orbit with radial distance $r = r(t)$ and azimuth $\phi = \phi(t)$ where $t$ denotes time. Distance and velocity at perigee are $R_{\mathrm{per}}$ and $V_{\mathrm{per}}$, respectively. The corresponding variables at apogee are $R_{\mathrm{apo}}$ and $V_{\mathrm{apo}}$, the gravitational constant is $G$, and the planetary mass is $M$. Combining the conservation laws for angular momentum

$$r^2 \dot{\phi} = R_{\mathrm{apo}} V_{\mathrm{apo}} = R_{\mathrm{per}} V_{\mathrm{per}} \tag{C1}$$

and total energy $E$ (here normalized by the test mass $m$)

$$\frac{E}{m} = \frac{1}{2} \left( \dot{r}^2 + r^2 \dot{\phi}^2 \right) - \frac{GM}{r} \tag{C2}$$

$$= \frac{1}{2} V_{\mathrm{per}}^2 - \frac{GM}{R_{\mathrm{per}}} = \frac{1}{2} V_{\mathrm{apo}}^2 - \frac{GM}{R_{\mathrm{apo}}} \tag{C3}$$

yields the following expression for the perigee velocity in terms of perigee and apogee distances

$$V_{\mathrm{per}}^2 = \frac{2 G M R_{\mathrm{apo}}}{R_{\mathrm{per}}(R_{\mathrm{apo}} + R_{\mathrm{per}})} = \frac{2 g_{\mathrm{per}} R_{\mathrm{per}} R_{\mathrm{apo}}}{(R_{\mathrm{apo}} + R_{\mathrm{per}})} \tag{C4}$$

where $g_{per} = \frac{GM}{R_{per}^2}$ is the value of Earth's gravitational acceleration at geocentric distance $R_{per}$. The radial velocity $\dot{r}$ satisfies

$$\dot{r}^2 = \frac{2E}{m} + \frac{2GM}{r} - \frac{(r^2\dot{\phi})^2}{r^2} \tag{C5}$$

$$= \frac{2E}{m} + \frac{2GM}{r} - \frac{R_{per}^2 V_{per}^2}{r^2} \ . \tag{C6}$$

Differentiating this expression and dividing by $2\dot{r}$ yields

$$\ddot{r} = -\frac{GM}{r^2} + \frac{R_{per}^2 V_{per}^2}{r^3} \ . \tag{C7}$$

Evaluation at perigee $r = R_{per}$ gives

$$\ddot{r}|_{r=R_{per}} = -\frac{GM}{R_{per}^2} + \frac{R_{per}^2 V_{per}^2}{R_{per}^3} = -\frac{GM}{R_{per}^2} + \frac{V_{per}^2}{R_{per}} \ . \tag{C8}$$

Inserting the expression for $V_{per}^2$ yields

$$\ddot{r}|_{r=R_{per}} = \frac{GM}{R_{per}^2} \frac{R_{apo} - R_{per}}{R_{apo} + R_{per}} = g_{per}\, \varepsilon \tag{C9}$$

where $\varepsilon = \frac{R_{apo}-R_{per}}{R_{apo}+R_{per}}$ is the orbital eccentricity. The altitude $z$ is related to radial distance $r$ and the Earth's planetary radius $R_E$ through $z = r - R_E$. At perigee, $t = 0$ and $z = z_{per}$. The parameter $a_{per} = \ddot{z}(t=0)$ coincides with the radial acceleration at perigee $\ddot{r}|_{r=R_{per}}$. Hence, orbital altitudes around perigee are approximately given by the quadratic function

$$z(t) \simeq z_{per} + \frac{a_{per}}{2}t^2 \ . \tag{C10}$$

To the same approximation order, the angular momentum conservation condition $r^2\dot{\phi} = R_{per}V_{per}$ can be integrated to yield

approximate azimuths $\phi = \phi(t)$. In $\dot{\phi} = R_{per}V_{per}/r^2$ insert $r = r(t) = R_{per} + \frac{a_{per}}{2}t^2$, then expand

$$\left(R_{per} + \frac{a_{per}}{2}t^2\right)^{-2} \simeq R_{per}^{-2}\left(1 - \frac{a_{per}}{R_{per}}t^2\right) \tag{C11}$$

and integrate $d\phi = R_{per}V_{per}r^{-2}dt$ to obtain

$$\phi(t) \simeq \frac{V_{per}}{R_{per}}\int_0^t \left(1 - \frac{a_{per}}{R_{per}}\tilde{t}^2\right)d\tilde{t} \tag{C12}$$

$$= \frac{V_{per}}{R_{per}} \cdot \left(t - \frac{a_{per}}{3R_{per}}t^3\right) \ . \tag{C13}$$

The corresponding horizontal distances at the Earth's surface are then given by $x = x(t) = R_E\phi(t)$. By using Eq. (C9), this can be further processed to yield

$$x(t) \simeq \frac{R_E V_{per} t}{R_{per}} \cdot \left(1 - \frac{\varepsilon}{3}\frac{g_{per}t^2}{R_{per}}\right) \ . \tag{C14}$$

The leading term is ground distance for a circular orbit. The correction produced by the second term is proportional to eccentricity.

# Appendix D: Parametrization of horizontal electron density variations

In the initial version of the DIPCont package, the horizontal variability of electron density profiles is controlled by the keyword argument `LTIModelType`. Setting `LTIModelType='NeAuroralZoneCrossing'` produces two electron density maxima along the horizontal (latitudinal) axis as observed by a polar orbiting satellite when crossing the auroral oval, see Figures 1, 7, and 8. More specifically, the horizontal ($x$) variations of peak altitude $z_* = z_*(x)$ and peak electron density $N_{e*} = N_{e*}(x)$ in Eq. (15) are prescribed by the ad hoc parametrizations

$$z_*(x) \;=\; z_{*,\mathrm{min}} + \Delta z_* \cdot f(x)\,, \tag{D1}$$

$$N_{e*}(x) \;=\; N_{e*,\mathrm{max}} - \Delta N_{e*} \cdot f(x)\,, \tag{D2}$$

with

$$f(x) \;=\; \frac{1}{2}\left\{1 + \cos\left(\frac{4\pi x}{x_\mathrm{R} - x_\mathrm{L}}\right)\right\} \tag{D3}$$

so that $f = f(x)$ varies between zero and one. The parameters $x_\mathrm{L}$ and $x_\mathrm{R}$ are the horizontal boundaries of the modeling domain, here chosen to be $x_\mathrm{L} = -2000\,\mathrm{km}$ and $x_\mathrm{R} = 2000\,\mathrm{km}$. The values of the electron density peak parameters used in the model runs leading to Figures 1, 7, 8 are as follows: $z_{*,\mathrm{min}} = 110\,\mathrm{km}$, $\Delta z_* = 10\,\mathrm{km}$, $N_{e*,\mathrm{max}} = 1.5\cdot 10^{11}\,\mathrm{m}^{-3}$, $\Delta N_{e*} = 0.5\cdot 10^{11}\,\mathrm{m}^{-3}$.

All LTI model parameters for the simulation runs of the current report, including the horizontal electron density profile parameters, are provided in the configuration file `DIPContBas.py` as part of the supplementary material.

*Author contributions.* The DIPCont project emerged from meetings of the SIFACIT project team (OM, AB, JV, NS) with the Daedalus/MASE workgroup at the Democritus University of Thrace (TS, ST, TB, DB, PP). JV developed the DIPCont methodology and the underlying theory, wrote the Python code and the first draft of the manuscript, and coordinated the writing of the paper. OM initiated LTI profile extrapolation under Daedalus Phase-0 science study, followed on by the activity of the SIFACIT team, and contributed text to Sections 1, 4, 5, 6. AB, LP, NS, and OM tested the DIPCont code and provided feedback for its improvement. SB, AB, OM, and LP checked the theory and the model derivations. TS, ST, TB, DB, PP contributed to early work on altitude reconstruction using simulation as well as rocket data, and the integration with MASE. All authors read and approved the final manuscript.

*Competing interests.* The authors declare that they have no conflict of interest.

*Acknowledgements.* Preliminary work on reconstruction of conductivity and Joule heating vertical profiles was carried on under Daedalus Phase-0 Science Study. The development of DIPCont was supported by the SIFACIT project, ESA contract 4000118383.

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
