# Peer review of "Daedalus Ionospheric Profile Continuation (DIPCont): Monte Carlo Studies Assessing the Quality of In Situ Measurement Extrapolation"

_Geoscientific Instrumentation, Methods and Data Systems, 2022_

## Author Comment (AC1)

**Daedalus Ionospheric Profile Continuation (DIPCont): Monte Carlo Studies Assessing the Quality of In Situ Measurement Extrapolation — Reply to Reviewer 1 —**

Joachim Vogt[1,7], Octav Marghitu[2], Adrian Blagau[2,1,7], Leonie Pick[3,1,7], Nele Stachlys[4,1,7], Stephan Buchert[5], Theodoros Sarris[6], Stelios Tourgaidis[6], Thanasis Balafoutis[6], Dimitrios Baloukidis[6], and Panagiotis Pirnaris[6]

[1]School of Science, Constructor University, Campus Ring, 28759 Bremen, Germany
[2]Institute for Space Science, Str. Atomistilor 409, Ro 077125, Bucharest-Magurele, Romania
[3]Institute for Solar-Terrestrial Physics, German Aerospace Center, Kalkhorstweg 53, 17235 Neustrelitz, Germany
[4]Leibniz Institute for Astrophysics Potsdam (AIP), An der Sternwarte 16, 14482 Potsdam, Germany
[5]Swedish Institute of Space Physics, Uppsala, 75121, Sweden
[6]Department of Electrical and Computer Engineering, Democritus University of Thrace, Xanthi, 67132, Greece
[7]Until December 2022, Constructor University operated under the name Jacobs University Bremen

**Correspondence:** Joachim Vogt (jvogt@constructor.university)

**Reply to Reviewer 1**

The authors thank both reviewers for carefully evaluating our manuscript, and for their valuable suggestions. The paper was amended and corrected in several ways detailed below. Abstract and Introduction (Section 1) were rewritten to clarify the objectives and the organization of the manuscript. In the revised version, it is emphasized that the focus of the study is on

5 the assessment of downward continuation (extrapolation) *quality* rather than the construction of a new parametric model of selected LTI variables. The LTI model presented in Section 2 is designed to demonstrate, illustrate, and test the probabilistic DIPCont framework, and the expectations towards that model are made explicit at the beginning of Section 2. The first two subsections of Section 2 were swapped to explain the LTI model setup further in the presentation of scale height parameters.

Below are our responses to the comments of the first reviewer Alessio Pignalberi.

10     In the manuscript I did not find any clear information about the magnetic latitudes the mission is going to cover, or better, which are the latitudes for which the calculations developed here are valid. From the figures shown in the manuscript and from the discussion, I suppose that the main goal is the polar/auroral latitudes where the Pedersen conductivity is of utmost importance at LTI altitudes, but this is not clearly stated in the manuscript. If so, this should be clearly stated in the introduction.

15     I wonder if the Daedalus orbit configuration will make possible to get data at low latitudes, and also to estimate the Hall conductivity.

The following text was included in the revised version of the Introduction.

Daedalus aims to perform in situ measurements in the LTI from an elliptical orbit, with a nominal perigee of 150 km and an apogee on the order of 2000 km. Very low altitudes down to 120 km will be sampled by use of propulsion, through a series of

20  short excursions in the form of perigee descent maneuvers. These are planned to be performed at high latitudes (>65 degrees magnetic latitude), where Pedersen conductivity and Joule heating maximize. The highly elliptical orbit of Daedalus leads to a natural precession of the orbit's semi major axis, both in magnetic latitude and in magnetic local time; this means that Daedalus will perform measurements along its elliptical orbit down to the nominal perigee of 150 km throughout all magnetic latitudes. The geophysical observables sampled by Daedalus will enable obtaining a series of derived products, as described in Table 1 of

25  the Daedalus Report for Assessment (ESA, 2020), which, among many others, include the calculation of Pedersen conductivity and Hall conductivity.

Line 57: About "and disregarding the contribution from electron-neutral collisions", please provide a reference to support this hypothesis or, alternatively, provide a numerical example.

The following text was included in the revised version of Section 2.

30  As explained in reviews of ionospheric physics (e.g., Rishbeth, 1997), contributions from electron-neutral collisions peak in the D-region but are unimportant at higher altitudes, see also Figure 4 in Sarris et al. (2023b).

Line 87: About "Disregarding altitude changes of atmospheric composition", ?I wonder how much the hypothesis of disregarding altitude changes of atmospheric composition could impact on the derivation of the neutral scale height vertical gradient. In fact, as also the authors explained before, in the LTI the atmosphere is not uniform

35  in composition and every constituent obeys to its own barometric law. The hypothesis made here seems to be in contrast with what has been said before. To substantiate your working hypothesis, I would suggest to verify the range of its applicability through the NRLMSISE-00 model.

The empirical atmospheric model NRLMSIS 2.0 was run for different seasons and a range of latitudes, to produce profiles of neutral LTI variables that are displayed in the supplementary figures S1a-S1d. The following text was included in the revised

40  version of Subsection 2.1.

Variations of gravity $g$ across the LTI are in the range of a few percent and can be neglected in this context. Profiles of $T_n$, $M_n$, and $H_n^P$ as predicted by the empirical atmospheric model NRLMSIS 2.0 (Emmert et al., 2021) for different seasons and latitudes are displayed in Figures S1a–S1d as part of the supplementary material to this paper, indicating that relative variations of average molar mass are indeed significantly smaller than those of neutral temperature. We thus disregard altitude changes in

45  average molar mass $M_n$ as imposed by changes in atmospheric composition, and further assume that temperature $T_n$, pressure scale height $H_n^P$, and density $H_n^N$ vary linearly with altitude in a self-consistent manner as described by Eqs. (4) and (5).

Line 152: About "For simplicity, the ion gyrofrequency is set to a constant.", I suppose constant with the respect to the altitudinal variation once the location is set, isn't it?

The International Reference Model (IRI) 2020 was run for different seasons and a range of latitudes, to produce the supple-

50  mentary figures S1a-S1d. The following text was included in the revised version of Subsection 2.7.

Furthermore, in the logic of the LTI model constructed for the initial version of the DIPCont package, changes in atmospheric composition and thus average ion mass are disregarded. Inspection of Figures S2a–S2d in the supplementary material to this report indicate that in the lower part of the LTI (altitudes below about 150 km) being the focus of downward continuation quality in the current study, variations of average ion mass with altitude are relatively small. Hence, altitude variations of ion gyrofrequency are neglected. In the same way as for other LTI model variables, namely, through the dependence of the parameters in the vector $\mathbf{p} = \mathbf{p}(x)$ (see Subsection 2.8 below) on the coordinate $x$, horizontal variations of magnetic field strength $B$ and thus ion gyrofrequency $\Omega_i$ can be modeled.

Lines 304-306: About "electron density makes the main contribution to the peaked height variation of Pedersen conductivity...", this is true but, to convince a skeptical reader about this, I would present also the plots for the neutral density, ion temperature and ion-collision frequency for the case shown in Figures 4-6. It is enough to show vertical profiles like Figure 5. These plots would also make clearer the altitudinal variations of these parameters as defined by the equations derived in the paper, and could be useful for the discussion of the results.

Supplementary Figure S3 contains additional information on the DIPCont model run producing the results in Figures 4, 5, 6. The following text was included in the revised version of Subsection 3.4.

Figure S3 in the supplementary material to this report provides additional information on this DIPCont model run, visualizing model distributions, ensembles of altitude profiles, and extrapolation horizons also for neutral temperature $T_n$, neutral density $N_n$, ion temperature $T_i$, and ion-neutral collision frequency $\nu_{in}$.

Lines 306-307: About "Pedersen conductivity controls the height variation of Joule heating", I would show the analytical dependence between these two parameters. Adding another equation to the paper should not be a problem given the number of equations already present.

The following text was included in the revised version of Section 4.

In the neutral wind reference frame, Joule heating is $\mathbf{j}_\perp \cdot \mathbf{E}_\perp = \sigma_P |\mathbf{E}_\perp|^2$ where the subscript $\perp$ indicates a vectorial component perpendicular to the ambient magnetic field direction $\hat{\mathbf{B}}$. Height variations of $\mathbf{E}_\perp$ are negligible according to the following rationale, see, e.g., (Rishbeth, 1997). Due to high parallel conductivity, the electric field component $E_\parallel = E_s$ parallel to $\hat{\mathbf{B}}$ vanishes, i.e., $0 = E_s = -\frac{\partial \Phi}{\partial s}$, where $s$ is the magnetic field line coordinate, and $\Phi$ denotes the electric potential. The electric field component $E_q$ in a direction perpendicular to $\hat{\mathbf{B}}$ captured by a coordinate $q$ then satisfies $\frac{\partial E_q}{\partial s} = -\frac{\partial}{\partial s}\frac{\partial \Phi}{\partial q} = -\frac{\partial}{\partial q}\frac{\partial \Phi}{\partial s} = \frac{\partial E_s}{\partial q} = 0$.

Lines 317-317: About "In Figure 1 and in the following, latitudinal inhomogeneity of electron density....", is the crossing of the auroral oval taken just as an example or will be constrained by the orbit configuration?

The Daedalus orbit configuration is characterized in the revised version of the Introduction. Regarding the horizontal variation of electron density in Figures 1, 7, 8, the following text was included in the revised version of Section 4.

Since the physics of energetic particle precipitation is not incorporated in this initial version of the DIPCont package, the horizontal variation of electron density expected for an auroral oval crossing is prescribed through ad hoc choices of horizontal

electron density peak parameters profiles, see the option `LTIModelType='NeAuroralZoneCrossing'` in the DIPCont code as part of the supplementary material to this report.

85      Lines 371-372: About "The DIPCont package contains a parameter to study the effect of F-layer residuals on...", probably, the dayside F1 layer might slightly affect the electron density in the range 150-200 km of altitude, above all in the summer season. This is a point to check in future as a function of the perigee altitude.

Thanks for the suggestion.

Lines 410-414: In my opinion, this part is not very clear as it is written. Indeed, the derivation of (A4) on the base
90      of (A3) is based on the fact that $d \ln N_n = -dz/H_n^N$ which in turn leads to (A5). As a consequence, in my view, is the adoption of $d \ln N_n = -dz/H_n^N$ who leads to (A4) and not vice versa. I am not questioning the correctness of this part but only the way in which it is presented. Moreover, it should be make clearer the difference between the pressure scale height and the density scale height. Line 436: About "In the isothermal limit...", as a consequence, $H^N$ tells us how the scale height $H^P$ changes for a non-isothermal atmosphere. This will solve my previous
95      comment regarding the relation between $H^N$ and $H^P$, and should be put in evidence in the text.

The respective paragraphs were rewritten as follows.
Rearranging $-\frac{dz}{H_n^P} = \frac{dP_n}{P_n} = d \ln P_n$ and integrating leads to

$$P_n(z) = P_{n0} \exp \left\{ -\int_{z_0}^{z} \frac{d\tilde{z}}{H_n^P(\tilde{z})} \right\}$$

where the altitude dependence of $H_n^P$ directly reflects the change of temperature $T_n$ with $z$.
Analogous differential and integral expressions for the neutral density, namely, $d \ln N_n = -\frac{dz}{H_n^N}$ and

$$N_n(z) = N_{n0} \exp \left\{ -\int_{z_0}^{z} \frac{d\tilde{z}}{H_n^N(\tilde{z})} \right\} \, ,$$

are derived as follows. Combining the differential of the ideal gas law $dP_n = N_n k \, dT_n + k T_n \, dN_n$ with the hydrostatic condition yields $-N_n m_n g \, dz = N_n k \, dT_n + k T_n \, dN_n$ and thus $-\frac{m_n g}{kT_n} dz - \frac{1}{T_n} dT_n = \frac{1}{N_n} dN_n = d \ln N_n$. Since $\frac{dT_n}{T_n} = d \ln T_n = d \ln H^P = \frac{dH_n^P}{H_n^P}$, one obtains

$$\frac{d \ln N_n}{dz} = -\frac{1}{H_n^P} \left( 1 + \frac{dH_n^P}{dz} \right) \, .$$

Therefore, the density scale height $H_n^N$ in the expression $d \ln N_n = -\frac{dz}{H_n^N}$ is given by

$$H_n^N = H_n^P \left( 1 + \frac{dH_n^P}{dz} \right)^{-1} \, .$$

Line 438: About "Following the approach first presented by Chapman (1931),", Your derivation is based on the assumption of a single atmospheric constituent, like in the Chapman original derivation. Have you verified the reliability of this assumption in the LTI region and in the formation of the E layer? I suppose that the E layer should be the superposition of Chapman-like layers from O2+, N2+ and NO+ ions. This point should be at least discussed.

In the revised version of the manuscript, the single-constituent assumption is discussed in detail in Section 2, supported by model runs of the empirical models NRLMSIS 2.0 and IRI 2020, with results shown in supplementary figures S1a–S1d and S2a–S2d. Specifically, concerns regarding the multi-ion composition in the lower part of the LTI are addressed in the revised version of Subsection 2.7 as follows.

Furthermore, in the logic of the LTI model constructed for the initial version of the DIPCont package, changes in atmospheric composition and thus average ion mass are disregarded. Inspection of Figures S2a–S2d in the supplementary material to this report indicate that in the lower part of the LTI (altitudes below about 150 km) being the focus of downward continuation quality in the current study, variations of average ion mass with altitude are relatively small. Hence, altitude variations of ion gyrofrequency are neglected.

Appendix B: From the equations in Appendix B, I suppose that the $z$ axis has been taken increasing towards the ground. Otherwise, the minus sign should appear in (B1) and in the following equations in the exponential. In my view, this choice is not the best one because it does not make clear that the radiation is absorbed by neutral particles through the radiation path. Anyway, the direction of the $z$ axis should be clearly stated in the text.

In Appendix B, the $z$ axis points upwards. Radiation enters from above, energy is absorbed between altitudes $z + dz$ and $z$, the intensity change $dI = I(z + dz) - I(dz)$ is positive. Radiation intensity decreases along its path down the atmosphere, hence it must increase with altitude, and $dI/dz > 0$ as given by (B1). The direction of the $z$ axis has been made explicit in Appendix B through the following addition.

Here $z$ is altitude, and the $z$ axis is pointing upwards as before.

Line 84: Suggestion about the use of P for the scale height. Many people working in the ionosphere field could confuse it with the plasma scale height because of the presence of P.

Thanks for alerting us to this potential source of confusion. The symbol P is further overused in this context as it also indicates Pedersen conductivity and Pedersen currents. Since we could not think of another symbol that could equally well indicate pressure, however, we kept the notation.

Line 366: controling –> controlling

Corrected.

Line 367: the the –> the

130    Corrected.

      Line 442: precipitaion –> precipitation

Corrected.

      Eq. (B7) is just a repetition of Eq. (A15), it is not necessary to repeat it.

This is correct, but we decided to keep the repetition to facilitate the reading of the integral in (B8).

135          Line 508: aopgee –> apogee

Corrected.

**Daedalus Ionospheric Profile Continuation (DIPCont): Monte Carlo Studies Assessing the Quality of In Situ Measurement Extrapolation — Supplementary Figures S1a–S1d —**

Joachim Vogt[1,7], Octav Marghitu[2], Adrian Blagau[2,1,7], Leonie Pick[3,1,7], Nele Stachlys[4,1,7],
Stephan Buchert[5], Theodoros Sarris[6], Stelios Tourgaidis[6], Thanasis Balafoutis[6], Dimitrios Baloukidis[6],
and Panagiotis Pirnaris[6]

[1]School of Science, Constructor University, Campus Ring, 28759 Bremen, Germany
[2]Institute for Space Science, Str. Atomistilor 409, Ro 077125, Bucharest-Magurele, Romania
[3]Institute for Solar-Terrestrial Physics, German Aerospace Center, Kalkhorstweg 53, 17235 Neustrelitz, Germany
[4]Leibniz Institute for Astrophysics Potsdam (AIP), An der Sternwarte 16, 14482 Potsdam, Germany
[5]Swedish Institute of Space Physics, Uppsala, 75121, Sweden
[6]Department of Electrical and Computer Engineering, Democritus University of Thrace, Xanthi, 67132, Greece
[7]Until December 2022, Constructor University operated under the name Jacobs University Bremen

**Correspondence:** Joachim Vogt (jvogt@constructor.university)

**Supplementary Figures S1a–S1d**

The four diagrams show profiles of atmospheric density and temperature parameters computed from output of the empirical atmospheric model NRLMSIS 2.0 for 12:00 UT, geographic longitude $0°$, and three latitudes: $\beta = 15°$ (first row), $\beta = 45°$ (second row), $\beta = 75°$ (third row). Simulation results have been provided by the Community Coordinated Modeling Center at

5  Goddard Space Flight Center through their publicly available simulation services (https://ccmc.gsfc.nasa.gov). The empirical atmospheric model NRLMSIS 2.0 was developed by John Emmert and Douglas Drob at NRL. For further information, see

> Emmert, J. T., Drob, D. P., Picone, J. M., Siskind, D. E., Jones, M. Jr., Mlynczak, M. G., et al. (2020). *NRLMSIS 2.0: A whole-atmosphere empirical model of temperature and neutral species densities.* Earth and Space Science, 7, e2020EA001321. https://doi.org/ 10.1029/2020EA001321

10  The first column displays the total mass density together with partial mass densities of $N_2$, $O_2$, O, and Ar. The second column shows the average neutral molar mass $\langle M_n \rangle$ in units of g/mol as given by $\langle M_n \rangle = \sum_s M_{n,s} N_{n,s} / \sum_s N_{n,s}$, where $M_{n,s}$ is the molar mass of species $s$. The corresponding number densities $N_{n,s}$ are provided by the NRLMSIS 2.0 model, as well as the neutral temperature profile shown in the third column. In the fourth column, the resulting profle of pressure scale height is shown. The fifth column displays the relative change of neutral temperature and of average neutral molar mass. Each diagram

15  represent one of the four seasons in the year 2018: Figure S1a – Spring equinox, 20 March 2018; Figure S1b – Summer solstice, 21 June 2018; Figure S1c – Autumn equinox, 23 September 2018; Figure S1d – Winter solstice, 21 December 2018.

**Figure S1a (Spring equinox, 20 March 2018, 12:00 UT, geographic longitude 0°)**

[Figure]

**Figure S1b (Summer solstice, 21 June 2018, 12:00 UT, geographic longitude 0°)**

[Figure]

**Figure S1c (Autumn equinox, 23 September 2018, 12:00 UT, geographic longitude 0°)**

[Figure]

**Figure S1d (Winter solstice, 21 December 2018, 12:00 UT, geographic longitude 0°)**

[Figure]

**Daedalus Ionospheric Profile Continuation (DIPCont): Monte Carlo Studies Assessing the Quality of In Situ Measurement Extrapolation — Supplementary Figures S2a–S2d —**

Joachim Vogt[1,7], Octav Marghitu[2], Adrian Blagau[2,1,7], Leonie Pick[3,1,7], Nele Stachlys[4,1,7],
Stephan Buchert[5], Theodoros Sarris[6], Stelios Tourgaidis[6], Thanasis Balafoutis[6], Dimitrios Baloukidis[6],
and Panagiotis Pirnaris[6]

[1]School of Science, Constructor University, Campus Ring, 28759 Bremen, Germany
[2]Institute for Space Science, Str. Atomistilor 409, Ro 077125, Bucharest-Magurele, Romania
[3]Institute for Solar-Terrestrial Physics, German Aerospace Center, Kalkhorstweg 53, 17235 Neustrelitz, Germany
[4]Leibniz Institute for Astrophysics Potsdam (AIP), An der Sternwarte 16, 14482 Potsdam, Germany
[5]Swedish Institute of Space Physics, Uppsala, 75121, Sweden
[6]Department of Electrical and Computer Engineering, Democritus University of Thrace, Xanthi, 67132, Greece
[7]Until December 2022, Constructor University operated under the name Jacobs University Bremen

**Correspondence:** Joachim Vogt (jvogt@constructor.university)

**Supplementary Figures S2a–S2d**

The four diagrams show profiles of ionospheric parameters computed from output of the Ionospheric Reference Ionosphere (IRI) 2020 model for 12:00 UT, geographic longitude $0°$, and three latitudes: $\beta = 15°$ (first row), $\beta = 45°$ (second row), $\beta = 75°$ (third row). Simulation results have been provided by the Community Coordinated Modeling Center at Goddard Space

5 Flight Center through their publicly available simulation services (https://ccmc.gsfc.nasa.gov). The International Reference Ionosphere (IRI) 2020 Model was developed by the URSI/COSPAR Working Group on IRI. For further information, see

> Bilitza, D., Pezzopane, M., Truhlik, V., Altadill, D., Reinisch, B. W., Pignalberi, A. (2022). *The International Reference Ionosphere model: A review and description of an ionospheric benchmark.* Reviews of Geophysics, 60, e2022RG000792. https://doi. org/10.1029/2022RG000792

10     The first column displays the percentages $P_s$ of the three main contributor species $s$ to ion density in the region below 200 km, namely, $NO^+$, $O_2^+$, and $O^+$ ions. The second column shows the average ion molar mass $\langle M_i \rangle$ in units of g/mol as given by $\langle M_i \rangle = \sum_s M_{i,s} P_s / \sum_s P^s$, where $M_{i,s}$ is the molar mass of species $s$. Profiles of neutral temperature $T_n$, ion temperature $T_i$, and electron temperature $T_e$ are shown in the third column. Note that a dashed linestyle was chosen for the $T_i$ profile to show that in the range between 100 km and 200 km, it coincides with the $T_n$ profile. Each diagram represent one of

15 the four seasons in the year 2018: Figure S2a – Spring equinox, 20 March 2018; Figure S2b – Summer solstice, 21 June 2018; Figure S2c – Autumn equinox, 23 September 2018; Figure S2d – Winter solstice, 21 December 2018.

**Figure S2a (Spring equinox, 20 March 2018, 12:00 UT, geographic longitude 0°)**

[Figure]

**Figure S2b (Summer solstice, 21 June 2018, 12:00 UT, geographic longitude 0°)**

[Figure]

**Figure S2c (Autumn equinox, 23 September 2018, 12:00 UT, geographic longitude 0°)**

[Figure]

**Figure S2d (Winter solstice, 21 December 2018, 12:00 UT, geographic longitude 0°)**

[Figure]

**Daedalus Ionospheric Profile Continuation (DIPCont): Monte Carlo Studies Assessing the Quality of In Situ Measurement Extrapolation — Supplementary Figure S3 —**

Joachim Vogt[1,7], Octav Marghitu[2], Adrian Blagau[2,1,7], Leonie Pick[3,1,7], Nele Stachlys[4,1,7], Stephan Buchert[5], Theodoros Sarris[6], Stelios Tourgaidis[6], Thanasis Balafoutis[6], Dimitrios Baloukidis[6], and Panagiotis Pirnaris[6]

[1]School of Science, Constructor University, Campus Ring, 28759 Bremen, Germany
[2]Institute for Space Science, Str. Atomistilor 409, Ro 077125, Bucharest-Magurele, Romania
[3]Institute for Solar-Terrestrial Physics, German Aerospace Center, Kalkhorstweg 53, 17235 Neustrelitz, Germany
[4]Leibniz Institute for Astrophysics Potsdam (AIP), An der Sternwarte 16, 14482 Potsdam, Germany
[5]Swedish Institute of Space Physics, Uppsala, 75121, Sweden
[6]Department of Electrical and Computer Engineering, Democritus University of Thrace, Xanthi, 67132, Greece
[7]Until December 2022, Constructor University operated under the name Jacobs University Bremen

**Correspondence:** Joachim Vogt (jvogt@constructor.university)

**Supplementary Figure S3**

The graphics provides additional information on the DIPCont modeling results shown in Figures 4, 5, 6. Variables displayed in Figure S3: neutral temperature (first row), neutral density (second row), electron density (third row), ion temperature (fourth row), ion-neutral collision frequency (fifth row), Pederson conductivity (sixth row).

5    First column: Model distributions of LTI variables. Synthetic measurements are produced along the two satellite orbits (white dashed lines). The parameters of vertical profiles are estimated using measurements within a window (white solid rectangle) around two locations in horizontal direction (blue and green dashed lines).

Second column and fourth column: Visualization of the ensemble of altitude profiles generated from the Monte Carlo distributions of model parameters. Shown are selected quantiles evaluated at the vertical grid of LTI altitudes. Second column: center

10   position (blue dashed line) indicated in the first column diagram. Fourth column: right position (green dashed line) indicated in the first column diagram.

Third column and fifth column: Solid lines (blue and green) give the relative root-mean-square deviations of Monte Carlo altitude profiles from the respective input model profiles. Vertical dotted and solid lines represent a set of chosen error levels, ranging from 0.5 % and 1 % (yellow) to 32% and 64% (magenta). The corresponding horizontal lines show the extrapolation

15   horizons indicating at which altitude the relative deviation equals the respective error level. Third column: center position (blue dashed line) indicated in the first column diagram. Fifth column: right position (green dashed line) indicated in the first column diagram.

**Figure S3**

[Figure]

**Daedalus Ionospheric Profile Continuation (DIPCont): Monte Carlo Studies Assessing the Quality of In Situ Measurement Extrapolation — Supplementary Figures S4a–S4b —**

Joachim Vogt[1,7], Octav Marghitu[2], Adrian Blagau[2,1,7], Leonie Pick[3,1,7], Nele Stachlys[4,1,7],
Stephan Buchert[5], Theodoros Sarris[6], Stelios Tourgaidis[6], Thanasis Balafoutis[6], Dimitrios Baloukidis[6],
and Panagiotis Pirnaris[6]

[1]School of Science, Constructor University, Campus Ring, 28759 Bremen, Germany
[2]Institute for Space Science, Str. Atomistilor 409, Ro 077125, Bucharest-Magurele, Romania
[3]Institute for Solar-Terrestrial Physics, German Aerospace Center, Kalkhorstweg 53, 17235 Neustrelitz, Germany
[4]Leibniz Institute for Astrophysics Potsdam (AIP), An der Sternwarte 16, 14482 Potsdam, Germany
[5]Swedish Institute of Space Physics, Uppsala, 75121, Sweden
[6]Department of Electrical and Computer Engineering, Democritus University of Thrace, Xanthi, 67132, Greece
[7]Until December 2022, Constructor University operated under the name Jacobs University Bremen

**Correspondence:** Joachim Vogt (jvogt@constructor.university)

**Supplementary Figure S4a**

The diagram illustrates how visit times of ground horizontal distances are expected to differ for the two satellites of a dual-spacecraft mission to the LTI, provided they share the same orbital plane, have identical semi-major axes, and pass through their perigees at the same time. In the example, satellite A is on a Keplerian orbit with perigee altitude $z_{\mathrm{per}}^A = 130\,\mathrm{km}$ and apogee altitude $z_{\mathrm{apo}}^A = 3000\,\mathrm{km}$. Perigee and apogee altitudes of satellite B are $z_{\mathrm{per}}^B = 150\,\mathrm{km}$ and $z_{\mathrm{apo}}^B = 2980\,\mathrm{km}$, respectively. The orbits are computed using Stoermer-Verlet integration of the equation of motion. Left panel: Satellite altitudes $z$ versus ground horizontal distance $x$. Center panel: Satellite visit times $t^A = t^A(x)$ and $t^B = t^B(x)$ at ground horizontal distance $x$. Right panel: Difference $\Delta t(x) = t^B(x) - t^A(x)$ of satellite visit times at ground horizontal distance $x$.

**Supplementary Figure S4b**

The diagram provides additional information on the satellite orbit representations implemented in the DIPCont package, using a satellite on a Keplerian orbit with perigee altitude $z_{\mathrm{per}} = 130\,\mathrm{km}$ and apogee altitude $z_{\mathrm{apo}} = 3000\,\mathrm{km}$ as an example. Compared are the results of Stoermer-Verlet integration and the local polynomial approximation constructed in the Appendix of the manuscript. Time and ground horizontal distance are centered at the perigee location. Upper left panel: Altitude $z$ versus time $t$. Lower left panel: Ground horizontal distance $x$ versus time $t$. Upper right panel: Altitude deviation of the local polynomial approximation relative to the result of the Stoermer-Verlet integration. Lower right panel: Ground horizontal distance deviation of the local polynomial approximation relative to the result of the Stoermer-Verlet integration.

**Figure S4a**

[Figure]

**Figure S4b**

[revised manuscript text omitted]

---

## Author Comment (AC2)

**Daedalus Ionospheric Profile Continuation (DIPCont): Monte Carlo Studies Assessing the Quality of In Situ Measurement Extrapolation — Reply to Reviewer 2 —**

Joachim Vogt[1,7], Octav Marghitu[2], Adrian Blagau[2,1,7], Leonie Pick[3,1,7], Nele Stachlys[4,1,7], Stephan Buchert[5], Theodoros Sarris[6], Stelios Tourgaidis[6], Thanasis Balafoutis[6], Dimitrios Baloukidis[6], and Panagiotis Pirnaris[6]

[1]School of Science, Constructor University, Campus Ring, 28759 Bremen, Germany
[2]Institute for Space Science, Str. Atomistilor 409, Ro 077125, Bucharest-Magurele, Romania
[3]Institute for Solar-Terrestrial Physics, German Aerospace Center, Kalkhorstweg 53, 17235 Neustrelitz, Germany
[4]Leibniz Institute for Astrophysics Potsdam (AIP), An der Sternwarte 16, 14482 Potsdam, Germany
[5]Swedish Institute of Space Physics, Uppsala, 75121, Sweden
[6]Department of Electrical and Computer Engineering, Democritus University of Thrace, Xanthi, 67132, Greece
[7]Until December 2022, Constructor University operated under the name Jacobs University Bremen

**Correspondence:** Joachim Vogt (jvogt@constructor.university)

**Reply to Reviewer 2**

The authors thank both reviewers for carefully evaluating our manuscript, and for their valuable suggestions. The paper was amended and corrected in several ways detailed below. Abstract and Introduction (Section 1) were rewritten to clarify the objectives and the organization of the manuscript. In the revised version, it is emphasized that the focus of the study is on the assessment of downward continuation (extrapolation) *quality* rather than the construction of a new parametric model of selected LTI variables. The LTI model presented in Section 2 is designed to demonstrate, illustrate, and test the probabilistic DIPCont framework, and the expectations towards that model are made explicit at the beginning of Section 2. The first two subsections of Section 2 were swapped to explain the LTI model setup further in the presentation of scale height parameters.

Below are our responses to the comments of the second reviewer.

Paper organization not clear

The paper's purpose and organization should be reviewed and clarified in the Introduction. What are the main goals of the paper? Is the main objective to show how in situ measurements would ultimately provide ionospheric profiles? A natural question is how many such measurements are needed to obtain a realistic profile. Again, the overall objectives of the study need to be clarified.

Several parts of the paper were rewritten in response to this comment, most notably Abstract, Introduction, and Section 2. It is now emphasized that the DIPCont project is primarily concerned mainly with assessing the *quality* of downward continuation of in situ measurements as reflected in probabilistic measures of deviation obtained through Monte Carlo simulations. The

LTI model developed in the Appendices and presented in Section 2 was designed to demonstrate the DIPCont setup and methodology.

20    For example, consider the following revised paragraph in the Introduction.

The DIPCont procedure to assess the quality of in situ measurement downward continuation is detailed in Section 3. In brief, after choosing a LTI model, [. . . ] It is important to note that the filled contour representations of electron density and Pedersen conductivity model distributions mainly serve to provide contextual information, while the essential results of the DIPCont modeling procedure are the extrapolation horizons represented as plain contour lines, in response to the satellite orbit

25    configuration (white lines). The extrapolation horizons of the model run shown in Figure 1 suggest that for a dual-satellite mission as anticipated in the Daedalus Report for Assessment (ESA, 2020), downward continuation yields relative errors of a few ten percent at altitudes where electron density and Pedersen conductivity maximizes. Implications are discussed in more detail further below in Section 4, and contrasted with the single-satellite case.

It appears as if the authors are considering mid-latitude daytime conditions. If so, this should be stated.

30    High latitude conditions with auroral input would completely change the approach of this paper, since the iono-spheric plasma density is highly variable due to precipitating, energetic (auroral) particles. (See Figure 43 of Pfaff et al., Space Science Reviews, 2012, for an illustration of how the thermal plasma might vary depending on the incoming auroral electron precipitation.) The Daedalus objectives suggest that high latitudes are a key region that that mission seeks to understand.

35    In the revised version of the manuscript, it is clearly stated that the parametric models included in the initial version of the DIPCont package provide a simplified description of LTI, choosing the process of Pedersen conductivity formation as an indicative example to demonstrate the procedure and key DIPCont products. To this end, a single-species LTI model with an intentionally limited set of parameters and an incomplete representation of ionization processes is constructed.

The following paragraph was added to the Introduction.

40    The LTI model used to introduce and demonstrate the DIPCont methodology in this paper is presented in Section 2. The parametric model captures the whole LTI temperature range and thus addresses a main source of variability. To limit the number of model parameters and thus also instabilities during model inversion in this initial DIPCont study, LTI variables showing less pronounced changes and ionization source mechanisms are treated in a simplified manner. Furthermore, since the quality of downward continuation is in the focus of our study, the LTI model is restricted to E-region physics, with the influence of the

45    F-region left for future work.

The following text was added to Section 2.

Probabilistic measures of extrapolation quality produced by the DIPCont procedure detailed in Section 3 are based on synthetic in situ observations predicted by a model of the LTI. As emphasized in space physics textbooks and reviews of the LTI (e.g., Pfaff, 2012; Richmond 1995) the full complexity of LTI variability and dynamics calls for a full multi-species

50    description, taking into account source and loss processes varying in importance and efficiency as functions of magnetic latitude and local time and further factors. In the future, DIPCont functionality is planned to be included in the Daedalus

MASE (Mission Assessment through Simulation Exercise) toolset (Sarris et al., 2023b), designed with the purpose to assess and demonstrate the closure of the mission objectives of the proposed Daedalus mission.

The more complex the LTI model of choice, however, the larger the number of parameters that are to be estimated with a downward continuation of in situ satellite measurements, which in turn tend to negatively affect the stability of model inversion. With these implications in mind, the initial version of the DIPCont package contains a simplified LTI description based on a limited set of parameters. Extrapolation quality of a single but important process, namely, the formation of Pedersen conductivity $\sigma_P$, is supposed to be studied in a self-consistent manner. To this end, only a single particle species is considered, and classical photoionization physics is applied to parametrize ionospheric layer formation. [. . . ]

The following text was added to Section 4.

Since the physics of energetic particle precipitation is not incorporated in this initial version of the DIPCont package, the horizontal variation of electron density expected for an auroral oval crossing is prescribed through ad hoc choices of horizontal electron density peak parameters profiles, see the option `LTIModelType='NeAuroralZoneCrossing'` in the DIPCont code as part of the supplementary material to this report.

Challenges with dual satellite investigation data

The use of two satellites to gather the profile data is a little difficult to follow. Because the satellites have different perigees, their orbital periods would be different. It is hard to believe that two satellites would gather data exactly simultaneously, as shown in numerous figures. How would the results differ if the two orbits were not synchronous or not in the same plane?

In the revised manuscript, dual-satellite orbit geometry is clarified and further explained.

The following text was included in the revised version of Subsection 3.1, see also Supplementary Figure S4a.

When dual-satellite missions to the LTI are considered, the question arises how synchronous the measurements are with respect to ground horizontal distance $x$, assuming the two spacecraft share the same orbital plane, have identical semi-major axes and thus orbital periods, and pass through their perigees at the same time. Figure S4a in the supplementary material to this report illustrates how visit times of ground horizontal distances are expected to differ for two satellites with perigee altitudes 130 km and 150 km. Differences of satellite visit times turn out to be on the order of seconds.

The following text was added to Subsection 4.1.

Note that in all dual-satellite DIPCont model runs presented in this paper, apogee distances of the second satellite have been adjusted such that the sum of perigee and apogee distances are identical for both satellites, and thus also the semi-major axes and the orbital periods.

Ions and other parameters not specified

The analysis discusses ion-neutral collisions, but the paper does not specify which ion species are used and which are the most common within the 100-200 km regime. The collision cross section value is given on page 7, so the lack of ion species specification is confusing.

85    Page 4, equation (1): only one ion and one neutral species are considered in the model. Clearly these two ion species are not the same at all altitudes, so this must be clarified. The collision cross section is given later but it is important to have some explanation of which ions are used.

Page 5, eq (4). It is surprising to have a constant Mn between 100-200 km since the ion mass changes with altitude. According to Appendix A, this mass represents the average mass, but this is not realistic since the collision

90    frequencies are different for different species.

Page 6, line 129. The linear variation of Ti is not realistic.

Page 7, line 144. What is the reference for the collision cross section and for which species is this valid?

It is understood that the single-species approach cannot offer a complete description of LTI structure and dynamics. As stated before, in this first DIPCont paper, Pedersen conductivity formation is the process selected to demonstrate the methodology

95    and data products, so a simplified LTI description was chosen. Nonetheless, several additional efforts are made to motivate the choices of model parameters.

Average mass Mn is discussed in the revised version of Subsection 2.1.

Variations of gravity $g$ across the LTI are in the range of a few percent and can be neglected in this context. Profiles of $T_n$, $M_n$, and $H_n^P$ as predicted by the empirical atmospheric model NRLMSIS 2.0 (Emmert et al., 2021) for different seasons and

100   latitudes are displayed in Figures S1a–S1d as part of the supplementary material to this paper, indicating that relative variations of average molar mass are indeed significantly smaller than those of neutral temperature. We thus disregard altitude changes in average molar mass $M_n$ as imposed by changes in atmospheric composition, and further assume that temperature $T_n$, pressure scale height $H_n^P$, and density $H_n^N$ vary linearly with altitude in a self-consistent manner as described by Eqs. (4) and (5).

Ion composition is discussed in the revised version of Subsection 2.7, see also Supplementary Figures S2a–S2d.

105   Furthermore, in the logic of the LTI model constructed for the initial version of the DIPCont package, changes in atmospheric composition and thus average ion mass are disregarded. Inspection of Figures S2a–S2d in the supplementary material to this report indicate that in the lower part of the LTI (altitudes below about 150 km) being the focus of downward continuation quality in the current study, variations of average ion mass with altitude are relatively small.

Following the modeling logic explained at the beginning of Section 2, variabilities of scale heights and temperatures are

110   approximated by linear profiles. As discussed in the revised version of Subsection 2.5 and shown through IRI 2020 predictions in Supplementary Figures S2a–S2d, the ion temperature is very close to the neutral temperature in the LTI between 100 km and 200 km.

Temperature profiles obtained by the International Reference Ionosphere (IRI) 2.0 model (Bilitza et al., 2022) indicate that ion and neutral temperatures are very similar throughout the LTI, see Figures S2a–S2d in the supplementary material to this

115   report.

Formulas for the ion-neutral collision frequency and the collision cross section were taken from the NRL Plasma Formulary (Huba, 2019).

Latitude and Local Time not specified for examples shown

Although the paper presents a generic case for the method development and validation, the reader needs to know the latitude, longitude, local time, etc. used for the analysis. Are the simulations for the equator or mid-latitudes? What is the local time? How would the results be different if the passes were at night or in the auroral zone?

To specify Daedalus mission parameters, the following text was included in the revised version of the Introduction.

Daedalus aims to perform in situ measurements in the LTI from an elliptical orbit, with a nominal perigee of 150 km and an apogee on the order of 2000 km. Very low altitudes down to 120 km will be sampled by use of propulsion, through a series of short excursions in the form of perigee descent maneuvers. These are planned to be performed at high latitudes (>65 degrees magnetic latitude), where Pedersen conductivity and Joule heating maximize. The highly elliptical orbit of Daedalus leads to a natural precession of the orbit's semi major axis, both in magnetic latitude and in magnetic local time; this means that Daedalus will perform measurements along its elliptical orbit down to the nominal perigee of 150 km throughout all magnetic latitudes. The geophysical observables sampled by Daedalus will enable obtaining a series of derived products, as described in Table 1 of the Daedalus Report for Assessment (ESA, 2020), which, among many others, include the calculation of Pedersen conductivity and Hall conductivity.

Temporal Variations

The paper presents a case for the method development and validation for static conditions. How does the method react to changes in the environment during a pass? In other words, how sensitive is the analysis to temporal variations? How long is a pass in the simulations shown?

Representations of orbital positions (altitudes and ground horizontal distances) can be found in Supplementary Figure S4b. The synchronicity of dual-satellite measurements is addressed in Supplementary Figure S4a, and discussed in Section 3.1.

General Concern with Figures

Figures 1, 7, and 8 are perplexing. Why are there two peaks of the density and Pedersen conductivity near 115 km at +1000 km and -1000 km? Presumably this is mid latitude, daytime, based on the Chapman layer discussion. Why not show continuous plasma density and Pedersen conductivity as in Figure 4?

Figures 5-8. It is not easy to understand the results of these figures, although they appear to be at the core of the paper?s objectives. For example, Figures 5 and 6 show Monte Carlo predictions. To what do the percentiles refer and what is the main result that the authors wish to show? This is not explained clearly in the text.

Figures 7-8 show the results of the method for two satellites and one satellite. What are the main results from these figures that the authors seek to convey? Presumably the overall goal is to show altitude profiles of the parameters obtained from the in situ measurements which might then be compared with the model. The results are not clear at all.

Figures 5 and 6 show how error measures are constructed from ensembles of Monte Carlo predictions along altitude profiles, whereas in Figures 1, 7, 8 the one-dimensional (altitude) information is integrated in a two-dimensional setup. The main

information in Figures 1, 7, 8 are the extrapolation horizons as quantified by the error contours. The model distribution (filled contours) are provided as contextual information. This is now clearly stated already in the Introduction.

It is important to note that the filled contour representations of electron density and Pedersen conductivity model distributions mainly serve to provide contextual information, while the essential results of the DIPCont modeling procedure are the extrapolation horizons represented as plain contour lines, in response to the satellite orbit configuration (white lines). The extrapolation horizons of the model run shown in Figure 1 suggest that for a dual-satellite mission as anticipated in the Daedalus Report for Assessment (ESA, 2020), downward continuation yields relative errors of a few ten percent at altitudes where electron density and Pedersen conductivity maximizes. Implications are discussed in more detail further below in Section 4, and contrasted with the single-satellite case.

Regarding the horizontal variation of electron density in Figures 1, 7, 8, the following text was included in the revised version of Section 4 (discussed already in the context of another comment).

Since the physics of energetic particle precipitation is not incorporated in this initial version of the DIPCont package, the horizontal variation of electron density expected for an auroral oval crossing is prescribed through ad hoc choices of horizontal electron density peak parameters profiles, see the option `LTIModelType='NeAuroralZoneCrossing'` in the DIPCont code as part of the supplementary material to this report.

Minor Comments:

The paper's title is very confusing. Why say "Continuation" in the title? A suggested title is simply: "Daedalus Ionospheric Profile Study". "Continuation" and "DIPCont" could be explained in the main text but should not be in the title of the paper.

Page 4, eq (3). On the left-hand side, T should be Tn. Same on line 103 (page 5). Suggest the authors check everywhere where T is used in place of Tn, Te, Ti.

The title has been amended by the following subtitle: *Monte Carlo Studies Assessing the Quality of In Situ Measurement Extrapolation*. One instance of $T$ was changed to $T_n$, the other was deleted. The term "continuation" is established in geophysical potential theory for a process of model extrapolation, possibly using boundary data, and thus very appropriate in the current context.

**Daedalus Ionospheric Profile Continuation (DIPCont): Monte Carlo Studies Assessing the Quality of In Situ Measurement Extrapolation — Supplementary Figures S1a–S1d —**

Joachim Vogt[1,7], Octav Marghitu[2], Adrian Blagau[2,1,7], Leonie Pick[3,1,7], Nele Stachlys[4,1,7],
Stephan Buchert[5], Theodoros Sarris[6], Stelios Tourgaidis[6], Thanasis Balafoutis[6], Dimitrios Baloukidis[6],
and Panagiotis Pirnaris[6]

[1]School of Science, Constructor University, Campus Ring, 28759 Bremen, Germany
[2]Institute for Space Science, Str. Atomistilor 409, Ro 077125, Bucharest-Magurele, Romania
[3]Institute for Solar-Terrestrial Physics, German Aerospace Center, Kalkhorstweg 53, 17235 Neustrelitz, Germany
[4]Leibniz Institute for Astrophysics Potsdam (AIP), An der Sternwarte 16, 14482 Potsdam, Germany
[5]Swedish Institute of Space Physics, Uppsala, 75121, Sweden
[6]Department of Electrical and Computer Engineering, Democritus University of Thrace, Xanthi, 67132, Greece
[7]Until December 2022, Constructor University operated under the name Jacobs University Bremen

**Correspondence:** Joachim Vogt (jvogt@constructor.university)

**Supplementary Figures S1a–S1d**

The four diagrams show profiles of atmospheric density and temperature parameters computed from output of the empirical atmospheric model NRLMSIS 2.0 for 12:00 UT, geographic longitude $0°$, and three latitudes: $\beta = 15°$ (first row), $\beta = 45°$ (second row), $\beta = 75°$ (third row). Simulation results have been provided by the Community Coordinated Modeling Center at Goddard Space Flight Center through their publicly available simulation services (https://ccmc.gsfc.nasa.gov). The empirical atmospheric model NRLMSIS 2.0 was developed by John Emmert and Douglas Drob at NRL. For further information, see

> Emmert, J. T., Drob, D. P., Picone, J. M., Siskind, D. E., Jones, M. Jr., Mlynczak, M. G., et al. (2020). *NRLMSIS 2.0: A whole-atmosphere empirical model of temperature and neutral species densities.* Earth and Space Science, 7, e2020EA001321. https://doi.org/ 10.1029/2020EA001321

The first column displays the total mass density together with partial mass densities of $N_2$, $O_2$, O, and Ar. The second column shows the average neutral molar mass $\langle M_n \rangle$ in units of g/mol as given by $\langle M_n \rangle = \sum_s M_{n,s} N_{n,s} / \sum_s N_{n,s}$, where $M_{n,s}$ is the molar mass of species $s$. The corresponding number densities $N_{n,s}$ are provided by the NRLMSIS 2.0 model, as well as the neutral temperature profile shown in the third column. In the fourth column, the resulting profile of pressure scale height is shown. The fifth column displays the relative change of neutral temperature and of average neutral molar mass. Each diagram represent one of the four seasons in the year 2018: Figure S1a – Spring equinox, 20 March 2018; Figure S1b – Summer solstice, 21 June 2018; Figure S1c – Autumn equinox, 23 September 2018; Figure S1d – Winter solstice, 21 December 2018.

**Figure S1a (Spring equinox, 20 March 2018, 12:00 UT, geographic longitude 0°)**

[Figure]

**Figure S1b (Summer solstice, 21 June 2018, 12:00 UT, geographic longitude 0°)**

[Figure]

**Figure S1c (Autumn equinox, 23 September 2018, 12:00 UT, geographic longitude 0°)**

[Figure]

**Figure S1d (Winter solstice, 21 December 2018, 12:00 UT, geographic longitude 0°)**

[Figure]

**Daedalus Ionospheric Profile Continuation (DIPCont): Monte Carlo Studies Assessing the Quality of In Situ Measurement Extrapolation — Supplementary Figures S2a–S2d —**

Joachim Vogt[1,7], Octav Marghitu[2], Adrian Blagau[2,1,7], Leonie Pick[3,1,7], Nele Stachlys[4,1,7], Stephan Buchert[5], Theodoros Sarris[6], Stelios Tourgaidis[6], Thanasis Balafoutis[6], Dimitrios Baloukidis[6], and Panagiotis Pirnaris[6]

[1]School of Science, Constructor University, Campus Ring, 28759 Bremen, Germany
[2]Institute for Space Science, Str. Atomistilor 409, Ro 077125, Bucharest-Magurele, Romania
[3]Institute for Solar-Terrestrial Physics, German Aerospace Center, Kalkhorstweg 53, 17235 Neustrelitz, Germany
[4]Leibniz Institute for Astrophysics Potsdam (AIP), An der Sternwarte 16, 14482 Potsdam, Germany
[5]Swedish Institute of Space Physics, Uppsala, 75121, Sweden
[6]Department of Electrical and Computer Engineering, Democritus University of Thrace, Xanthi, 67132, Greece
[7]Until December 2022, Constructor University operated under the name Jacobs University Bremen

**Correspondence:** Joachim Vogt (jvogt@constructor.university)

**Supplementary Figures S2a–S2d**

The four diagrams show profiles of ionospheric parameters computed from output of the Ionospheric Reference Ionosphere (IRI) 2020 model for 12:00 UT, geographic longitude $0°$, and three latitudes: $\beta = 15°$ (first row), $\beta = 45°$ (second row), $\beta = 75°$ (third row). Simulation results have been provided by the Community Coordinated Modeling Center at Goddard Space

5   Flight Center through their publicly available simulation services (https://ccmc.gsfc.nasa.gov). The International Reference Ionosphere (IRI) 2020 Model was developed by the URSI/COSPAR Working Group on IRI. For further information, see

> Bilitza, D., Pezzopane, M., Truhlik, V., Altadill, D., Reinisch, B. W., Pignalberi, A. (2022). *The International Reference Ionosphere model: A review and description of an ionospheric benchmark.* Reviews of Geophysics, 60, e2022RG000792. https://doi. org/10.1029/2022RG000792

10   The first column displays the percentages $P_s$ of the three main contributor species $s$ to ion density in the region below 200 km, namely, $NO^+$, $O_2^+$, and $O^+$ ions. The second column shows the average ion molar mass $\langle M_i \rangle$ in units of g/mol as given by $\langle M_i \rangle = \sum_s M_{i,s} P_s / \sum_s P^s$, where $M_{i,s}$ is the molar mass of species $s$. Profiles of neutral temperature $T_n$, ion temperature $T_i$, and electron temperature $T_e$ are shown in the third column. Note that a dashed linestyle was chosen for the $T_i$ profile to show that in the range between 100 km and 200 km, it coincides with the $T_n$ profile. Each diagram represent one of

15   the four seasons in the year 2018: Figure S2a – Spring equinox, 20 March 2018; Figure S2b – Summer solstice, 21 June 2018; Figure S2c – Autumn equinox, 23 September 2018; Figure S2d – Winter solstice, 21 December 2018.

**Figure S2a (Spring equinox, 20 March 2018, 12:00 UT, geographic longitude 0°)**

[Figure]

**Figure S2b (Summer solstice, 21 June 2018, 12:00 UT, geographic longitude 0°)**

[Figure]

**Figure S2c (Autumn equinox, 23 September 2018, 12:00 UT, geographic longitude 0°)**

[Figure]

**Figure S2d (Winter solstice, 21 December 2018, 12:00 UT, geographic longitude 0°)**

[Figure]

**Daedalus Ionospheric Profile Continuation (DIPCont): Monte Carlo Studies Assessing the Quality of In Situ Measurement Extrapolation — Supplementary Figure S3 —**

Joachim Vogt[1,7], Octav Marghitu[2], Adrian Blagau[2,1,7], Leonie Pick[3,1,7], Nele Stachlys[4,1,7],
Stephan Buchert[5], Theodoros Sarris[6], Stelios Tourgaidis[6], Thanasis Balafoutis[6], Dimitrios Baloukidis[6],
and Panagiotis Pirnaris[6]

[1]School of Science, Constructor University, Campus Ring, 28759 Bremen, Germany
[2]Institute for Space Science, Str. Atomistilor 409, Ro 077125, Bucharest-Magurele, Romania
[3]Institute for Solar-Terrestrial Physics, German Aerospace Center, Kalkhorstweg 53, 17235 Neustrelitz, Germany
[4]Leibniz Institute for Astrophysics Potsdam (AIP), An der Sternwarte 16, 14482 Potsdam, Germany
[5]Swedish Institute of Space Physics, Uppsala, 75121, Sweden
[6]Department of Electrical and Computer Engineering, Democritus University of Thrace, Xanthi, 67132, Greece
[7]Until December 2022, Constructor University operated under the name Jacobs University Bremen

**Correspondence:** Joachim Vogt (jvogt@constructor.university)

**Supplementary Figure S3**

The graphics provides additional information on the DIPCont modeling results shown in Figures 4, 5, 6. Variables displayed in Figure S3: neutral temperature (first row), neutral density (second row), electron density (third row), ion temperature (fourth row), ion-neutral collision frequency (fifth row), Pederson conductivity (sixth row).

First column: Model distributions of LTI variables. Synthetic measurements are produced along the two satellite orbits (white dashed lines). The parameters of vertical profiles are estimated using measurements within a window (white solid rectangle) around two locations in horizontal direction (blue and green dashed lines).

Second column and fourth column: Visualization of the ensemble of altitude profiles generated from the Monte Carlo distributions of model parameters. Shown are selected quantiles evaluated at the vertical grid of LTI altitudes. Second column: center position (blue dashed line) indicated in the first column diagram. Fourth column: right position (green dashed line) indicated in the first column diagram.

Third column and fifth column: Solid lines (blue and green) give the relative root-mean-square deviations of Monte Carlo altitude profiles from the respective input model profiles. Vertical dotted and solid lines represent a set of chosen error levels, ranging from 0.5 % and 1 % (yellow) to 32% and 64% (magenta). The corresponding horizontal lines show the extrapolation horizons indicating at which altitude the relative deviation equals the respective error level. Third column: center position (blue dashed line) indicated in the first column diagram. Fifth column: right position (green dashed line) indicated in the first column diagram.

**Figure S3**

[Figure]

**Daedalus Ionospheric Profile Continuation (DIPCont): Monte Carlo Studies Assessing the Quality of In Situ Measurement Extrapolation — Supplementary Figures S4a–S4b —**

Joachim Vogt[1,7], Octav Marghitu[2], Adrian Blagau[2,1,7], Leonie Pick[3,1,7], Nele Stachlys[4,1,7],
Stephan Buchert[5], Theodoros Sarris[6], Stelios Tourgaidis[6], Thanasis Balafoutis[6], Dimitrios Baloukidis[6],
and Panagiotis Pirnaris[6]

[1]School of Science, Constructor University, Campus Ring, 28759 Bremen, Germany
[2]Institute for Space Science, Str. Atomistilor 409, Ro 077125, Bucharest-Magurele, Romania
[3]Institute for Solar-Terrestrial Physics, German Aerospace Center, Kalkhorstweg 53, 17235 Neustrelitz, Germany
[4]Leibniz Institute for Astrophysics Potsdam (AIP), An der Sternwarte 16, 14482 Potsdam, Germany
[5]Swedish Institute of Space Physics, Uppsala, 75121, Sweden
[6]Department of Electrical and Computer Engineering, Democritus University of Thrace, Xanthi, 67132, Greece
[7]Until December 2022, Constructor University operated under the name Jacobs University Bremen

**Correspondence:** Joachim Vogt (jvogt@constructor.university)

**Supplementary Figure S4a**

The diagram illustrates how visit times of ground horizontal distances are expected to differ for the two satellites of a dual-spacecraft mission to the LTI, provided they share the same orbital plane, have identical semi-major axes, and pass through their perigees at the same time. In the example, satellite A is on a Keplerian orbit with perigee altitude $z_{\mathrm{per}}^A = 130\,\mathrm{km}$ and apogee altitude $z_{\mathrm{apo}}^A = 3000\,\mathrm{km}$. Perigee and apogee altitudes of satellite B are $z_{\mathrm{per}}^B = 150\,\mathrm{km}$ and $z_{\mathrm{apo}}^B = 2980\,\mathrm{km}$, respectively. The orbits are computed using Stoermer-Verlet integration of the equation of motion. Left panel: Satellite altitudes $z$ versus ground horizontal distance $x$. Center panel: Satellite visit times $t^A = t^A(x)$ and $t^B = t^B(x)$ at ground horizontal distance $x$. Right panel: Difference $\Delta t(x) = t^B(x) - t^A(x)$ of satellite visit times at ground horizontal distance $x$.

**Supplementary Figure S4b**

The diagram provides additional information on the satellite orbit representations implemented in the DIPCont package, using a satellite on a Keplerian orbit with perigee altitude $z_{\mathrm{per}} = 130\,\mathrm{km}$ and apogee altitude $z_{\mathrm{apo}} = 3000\,\mathrm{km}$ as an example. Compared are the results of Stoermer-Verlet integration and the local polynomial approximation constructed in the Appendix of the manuscript. Time and ground horizontal distance are centered at the perigee location. Upper left panel: Altitude $z$ versus time $t$. Lower left panel: Ground horizontal distance $x$ versus time $t$. Upper right panel: Altitude deviation of the local polynomial approximation relative to the result of the Stoermer-Verlet integration. Lower right panel: Ground horizontal distance deviation of the local polynomial approximation relative to the result of the Stoermer-Verlet integration.

**Figure S4a**

[Figure]

**Figure S4b**

[revised manuscript text omitted]

---

## Author Response (AR2)

**Daedalus Ionospheric Profile Continuation (DIPCont): Monte Carlo Studies Assessing the Quality of In Situ Measurement Extrapolation — 2nd Revision : Reply to Reviewer 2 —**

Joachim Vogt[1,7], Octav Marghitu[2], Adrian Blagau[2,1,7], Leonie Pick[3,1,7], Nele Stachlys[4,1,7], Stephan Buchert[5], Theodoros Sarris[6], Stelios Tourgaidis[6], Thanasis Balafoutis[6], Dimitrios Baloukidis[6], and Panagiotis Pirnaris[6]

[1]School of Science, Constructor University, Campus Ring, 28759 Bremen, Germany
[2]Institute for Space Science, Str. Atomistilor 409, Ro 077125, Bucharest-Magurele, Romania
[3]Institute for Solar-Terrestrial Physics, German Aerospace Center, Kalkhorstweg 53, 17235 Neustrelitz, Germany
[4]Leibniz Institute for Astrophysics Potsdam (AIP), An der Sternwarte 16, 14482 Potsdam, Germany
[5]Swedish Institute of Space Physics, Uppsala, 75121, Sweden
[6]Department of Electrical and Computer Engineering, Democritus University of Thrace, Xanthi, 67132, Greece
[7]Until December 2022, Constructor University operated under the name Jacobs University Bremen

**Correspondence:** Joachim Vogt (jvogt@constructor.university)

**Second manuscript revision: Reply to Reviewer 2**

The authors thank both reviewers for carefully evaluating the revised manuscript.

Below are our responses to the comments of the second reviewer.

Comments on: *Daedalus Ionospheric Profile Continuation (DIPCont): Monte Carlo Studies Assessing the Quality of In Situ Measurement Extrapolation*, by Vogt et al.

This is a revised paper with a slightly different title that presents simulated calculations including extrapolations of altitude profiles of various ionospheric state parameters that would be measured by in situ instruments on a pair of low perigee, orbiting platforms such as the proposed Daedalus mission.

Although the paper is improved, there remains one area that the authors are asked to address. After this item is addressed, the paper should be ready for publication in Geospace Instrumentation.

1. Uncertainty of geophysical conditions corresponding to Figures 1, 7, and 8
This reviewer is still perplexed by Figures 1, 7, and 8 and the relevant parameters used for the analysis involving this figure.

First, it is not until the bottom of page 18 that the reader learns that the two peaks of the density and Pedersen conductivity near 115 km at +/-1000 km corresponds to simulated auroral precipitation. The reader also learns that the horizontal distance pertains to latitude and the center is the highest latitude, presumably in the polar cap. This should all be explained when Figure 1 is introduced and reflected in the caption.

The introductory text was amended as follows.

[...] The basic ideas are illustrated in Figure 1, displaying electron density and Pedersen conductivity extrapolation horizons for a range of relative error thresholds along the orbits of a dual-satellite mission. Horizontal distance corresponds to the latitudinal (north-south) direction, with the origin of the horizontal axis centered at the highest latitude along the satellite orbits. In the LTI model runs leading to Figure 1, latitudinal inhomogeneity parameters are set to reproduce the two electron density maxima observed by a polar orbiting satellite when crossing the auroral oval. See Section 4 and Appendix D for details. It is important to note that the filled contour representations of electron density and Pedersen conductivity model distributions mainly serve to provide contextual information, while the essential results of the DIPCont modeling procedure are the extrapolation horizons represented as plain contour lines, [...]

The caption of Figure 1 was amended as follows.

Extrapolation horizons and orbit configuration displayed on top of a two-dimensional section of the modeled LTI. Upper panel: Electron density $N_e$. Lower panel: Pedersen conductivity $\sigma_P$. Horizontal distance corresponds to the latitudinal (north-south) direction, with the origin of the horizontal axis centered at the highest latitude along the satellite orbits. In the LTI model runs leading to this figure, latitudinal inhomogeneity parameters are set to reproduce the two electron density maxima observed by a polar orbiting satellite when crossing the auroral oval. Synthetic measurements are produced [...]

The season and solar illumination used in the simulations is important for understanding the analysis particularly with respect to Figure 1, 7, 8. The reader needs to know what is the solar angle and understand the photoionization production for which the Chapman function is analyzed as part of the analysis. This should be clearly stated with respect to the parameters in Figure 1, 7, and 8.

As explained in our March 2023 response to the comments of reviewer 2, the DIPCont project is less concerned with LTI modeling but with assessing the *quality* of downward continuation of in situ measurements as reflected in probabilistic measures of deviation obtained through Monte Carlo simulations. In the March 2023 revision, major parts of the manuscript were rewritten to clarify the scope and the goals of the DIPCont project. The objective of this initial DIPCont paper is to introduce the probabilistic methodology and key concepts such as extrapolation horizons. The initial version of the DIPCont package contains a simplified description of the LTI, choosing the process of Pedersen conductivity formation as an indicative example to demonstrate the procedure and key DIPCont products. To this end, a single-species LTI model with an intentionally limited set of parameters and an incomplete representation of ionization processes is constructed.

At this stage, season and solar illumation are only implicitly represented through the set of (vertical) LTI model parameters listed in Table 1 and horizontal profile parameters that are available in the Python code as part of the supplementary material. The parameters used for the simulation runs in this initial DIPCont paper are the default parameters specified in the parameter configuration file `DIPContBas.py` as part of the supplementary material provided with this paper.

Now, as part of the second (August 2023) revision of the DIPCont paper, Appendix D is added to describe the simulation parameters and the functional form of the horizontal electron variations leading the LTI model setup in Figures 1, 7, and 8.

Appendix D: Parametrization of horizontal electron density variations

In the initial version of the DIPCont package, the horizontal variability of electron density profiles is controlled by the keyword argument `LTIModelType`. Setting `LTIModelType='NeAuroralZoneCrossing'` produces two electron density maxima along the horizontal (latitudinal) axis as observed by a polar orbiting satellite when crossing the auroral oval, see Figures 1, 7, and 8. More specifically, the horizontal ($x$) variations of peak altitude $z_* = z_*(x)$ and peak electron density $N_{e*} = N_{e*}(x)$ in Eq. (15) are prescribed by the ad hoc parametrizations

$$z_*(x) = z_{*,\mathrm{min}} + \Delta z_* \cdot f(x) \,, \tag{1}$$

$$N_{e*}(x) = N_{e*,\mathrm{max}} - \Delta N_{e*} \cdot f(x) \,, \tag{2}$$

with

$$f(x) = \frac{1}{2}\left\{1 + \cos\left(\frac{4\pi x}{x_{\mathrm{R}} - x_{\mathrm{L}}}\right)\right\} \tag{3}$$

so that $f = f(x)$ varies between zero and one. The parameters $x_{\mathrm{L}}$ and $x_{\mathrm{R}}$ are the horizontal boundaries of the modeling domain, here chosen to be $x_{\mathrm{L}} = -2000\,\mathrm{km}$ and $x_{\mathrm{R}} = 2000\,\mathrm{km}$. The values of the electron density peak parameters used in the model runs leading to Figures 1, 7, 8 are as follows: $z_{*,\mathrm{min}} = 110\,\mathrm{km}$, $\Delta z_* = 10\,\mathrm{km}$, $N_{e*,\mathrm{max}} = 1.5 \cdot 10^{11}\,\mathrm{m}^{-3}$, $\Delta N_{e*} = 0.5 \cdot 10^{11}\,\mathrm{m}^{-3}$.

All LTI model parameters for the simulation runs of the current report, including the horizontal electron density profile parameters, are provided in the configuration file `DIPContBas.py` as part of the supplementary material.

Furthermore, the third introductory paragraph to Section 4 was amended as follows.

Since the physics of energetic particle precipitation is not incorporated in this initial version of the DIPCont package, the horizontal variation of electron density expected for an auroral oval crossing is prescribed through ad hoc choices of horizontal electron density peak parameters profiles, see the option `LTIModelType='NeAuroralZoneCrossing'` in the DIPCont code as part of the supplementary material to this report. The functional forms of horizontal electron density peak parameters are given in Appendix D.

The authors use a constant magnetic field for all calculations referring to Figure 1 arguing that the gyro frequencies do not change noticeably in the 100 km of altitude under consideration. (This is surprising since it is very easy to accommodate the changing magnetic field in the analysis.) However, it is not clear if they are allowing the magnetic field to vary with latitude over the tens of degrees included in the simulation shown in Figure 1, 7, and 8. Surely the magnetic field would change over these distances corresponding to different latitudes, as this would influence gyro frequencies and the conductivities. Again, the authors should explain the geophysical circumstances in the calculations. This will help the reader understand the model calculations.

Yes, it would indeed be easy to accommodate the variability of the magnetic field, in both the vertical and the horizontal directions. However, as explained before, this initial version of the DIPCont model is meant to introduce and motivate the probabilistic methodology, and the number of model parameters is intentionally kept to a minimum. Since magnetic measurements

and models are significantly more accurate than other LTI observables, including a sophisticated magnetic field model at this stage would only increase the LTI model complexity but not contribute to assessing the most important sources of variability. In future work, variations of magnetic field strength are planned to be taken into account. The following text was added to the manuscript in Section 2.7.

[. . . ] In the same way as for other LTI model variables, namely, through the dependence of the parameters in the vector $\mathbf{p} = \mathbf{p}(x)$ (see Subsection 2.8 below) on the coordinate $x$, horizontal variations of magnetic field strength $B$ and thus ion gyrofrequency $\Omega_i$ can be modeled. , and are planned to be considered in future work.

Minor Comment:

2. Although the title refers to in situ measurements, the paper discusses only in situ measurements of state parameters, such as neutral and plasma density and neutral and plasma temperature. Other in situ measurements, for example of electric fields, neutral winds, currents, and energetic particles are not included in the study. Perhaps the title might say, *...in situ state parameter measurements...*

The DIPCont approach is sufficiently general to include electric fields and other observables besides state variables. The setup chosen to demonstrate the methodology in this initial DIPCont paper aims at a proof of concept rather than the complete picture, and thus intentionally concentrates on a limited number of observables and parameters to better control model complexity. It would be misleading to mention state variables so prominently, hence the authors prefer to abstain from further changes of the title.

Suggestion:

3. There are many other low perigee satellite studies in addition to Daedalus over the last 3 decades. It is suggested that the authors consider mentioning these in their Introduction as it would bolster the importance of the analysis and modeling results reported in their paper. These include the NASA TIMED mission which originally included 2 dipper missions, the NASA Geospace Electrodynamics Constellation that included 4 dippers (Grebowsky and Gervin, 2001), and numerous NASA Explorer missions similar to Daedalus, including the ASTRE Mission (Pfaff et al., 2022).

Thanks for the list of relevant mission proposals. In the new version of the manuscript, they have been included as follows. [. . . ] Since the early 20th century, the LTI has been studied extensively using ground-based remote sensing facilities such as ionosondes and radars, but in all aspects requiring in situ observations it remains underexplored territory. Rocket flights (e.g., Sangalli et al., 2009; Pfaff et al., 2022a) can offer only local and temporally confined information. Major technical challenges have so far prevented a satellite mission to the deep, dense part of the LTI, despite scientific interest, community proposals, and feasibility studies by major space agencies (e.g., Grebowsky and Gervin, 2001; Pfaff et al., 2022b). An early conception of the TIMED mission (e.g., Yee et al., 1999) considered dipper options for in situ investigations of the LTI. A recent initiative along this line is the Daedalus mission proposal (Sarris et al., 2020), submitted to ESA in response to the Explorer 10 Call [. . . ]

**References**

Grebowsky, J. M. and Gervin, J. C.: Geospace electrodynamic connections, Physics and Chemistry of the Earth C, 26, 253–258, https://doi.org/10.1016/S1464-1917(00)00117-3, 2001.

Pfaff, R., Kudeki, E., Freudenreich, H., Rowland, D., Larsen, M., and Klenzing, J.: Dual Sounding Rocket and C/NOFS Satellite Observations of DC Electric Fields and Plasma Density in the Equatorial E- and F-Region Ionosphere at Sunset, Journal of Geophysical Research (Space Physics), 127, e30191, https://doi.org/10.1029/2021JA030191, 2022a.

Pfaff, R., Rowland, D., Kepko, L., Benna, M., Heelis, R., Clemmons, J., and Thayer, J.: The Atmosphere-Space Transition Region Explorer (ASTRE) - Using in situ Measurements on a Low Perigee Satellite to Understand How the Upper Atmosphere and Magnetosphere are Coupled, in: 44th COSPAR Scientific Assembly. Held 16-24 July, vol. 44, p. 775, 2022b.

Yee, J.-H., Cameron, G. E., and Kusnierkiewicz, D. Y.: Overview of TIMED, in: Optical Spectroscopic Techniques and Instrumentation for Atmospheric and Space Research III, edited by Larar, A. M., vol. 3756, pp. 244 – 254, International Society for Optics and Photonics, SPIE, https://doi.org/10.1117/12.366378, 1999.